# A Unified Solution to Video Fusion: From Multi-Frame Learning to Benchmarking

**Zixiang Zhao**[1]    **Haowen Bai**[2]    **Bingxin Ke**[1]    **Yukun Cui**[2]
**Lilun Deng**[2]    **Yulun Zhang**[3]    **Kai Zhang**[4]    **Konrad Schindler**[1]

[1]ETH Zürich    [2]Xi'an Jiaotong University
[3]Shanghai Jiao Tong University    [4]Nanjing University

zixiang.zhao@ethz.ch

## Abstract

The real world is dynamic, yet most image fusion methods process static frames independently, ignoring temporal correlations in videos and leading to flickering and temporal inconsistency. To address this, we propose *Unified Video Fusion* (**UniVF**), a novel and unified framework for video fusion that leverages multi-frame learning and optical flow-based feature warping for informative, temporally coherent video fusion. To support its development, we also introduce *Video Fusion Benchmark* (**VF-Bench**), the first comprehensive benchmark covering four video fusion tasks: multi-exposure, multi-focus, infrared-visible, and medical fusion. VF-Bench provides high-quality, well-aligned video pairs obtained through synthetic data generation and rigorous curation from existing datasets, with a unified evaluation protocol that jointly assesses the spatial quality and temporal consistency of video fusion. Extensive experiments show that UniVF achieves state-of-the-art results across all tasks on VF-Bench. Project page: *vfbench.github.io*.

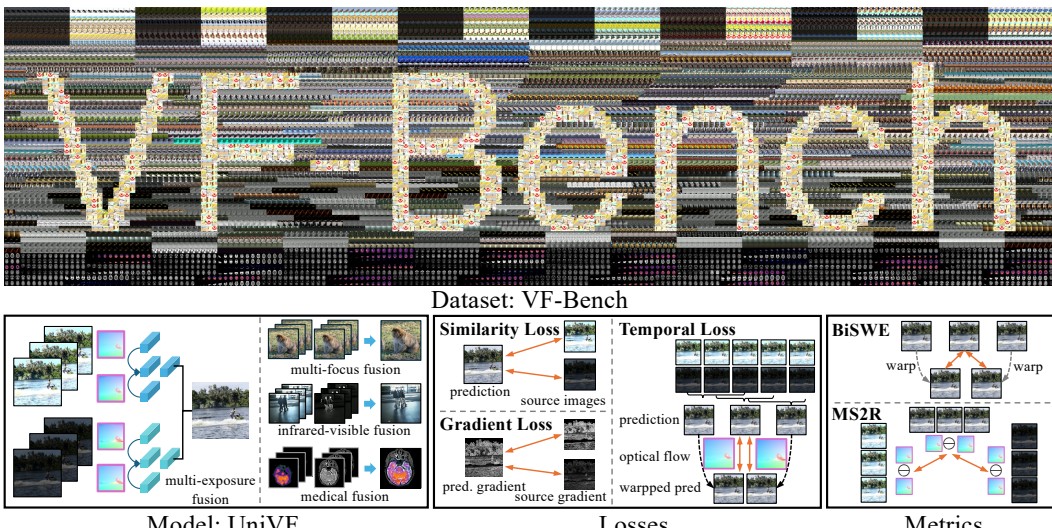

Figure 1: Overview of our main contribution in this paper.

## 1 Introduction

Image fusion has long been a key research direction in computer vision and image processing. It enables the combination of complementary information from multiple source images into a single,

39th Conference on Neural Information Processing Systems (NeurIPS 2025).

more informative and perceptually enhanced result [1–6]. A variety of fusion techniques, such as multi-exposure [7, 8], multi-focus [9, 4], infrared-visible [1, 2], and medical image fusion [10, 11], have proven valuable in applications, such as low-light enhancement and exposure correction [12, 13], extended depth-of-field imaging and generation of full-focus scenes [4], target recognition in adverse conditions [14], and improved diagnostic support in clinical imaging [15]. These approaches effectively overcome the limitations of individual sensors or imaging configurations, improving image interpretation by both human observers and machine vision systems. While image fusion has been extensively explored, transitioning to video fusion is a natural next step, as videos provide a continuous and temporally coherent view of dynamic scenes, including object and camera motion, transient events, and contextual variations [16]. With recently advanced imaging hardware and the increasing amount of video data, it has become feasible and necessary to extend image fusion to the temporal domain. The goal is to combine complementary information from multiple input videos into a single, temporally consistent output that offers a more complete representation of the scene.

However, the step from image to video domain introduces several new challenges beyond simply applying image fusion frame-by-frame: (i) *Leveraging temporal information*: Processing frames independently ignores the inherent temporal continuity of videos, leading to flickering and motion discontinuities. Effective video fusion must incorporate information from adjacent frames, not only improving per-frame quality but also ensuring temporal coherence. (ii) *Limited dataset scale*: Compared to paired images, collecting perfectly aligned, temporally synchronized, and diverse video pairs is a lot more challenging and expensive, limiting benchmarking and development for data-driven fusion approaches. (iii) *Lack of evaluation protocols*: Existing evaluation metrics are designed for images, while ignoring consistency along the temporal axis.

To tackle these challenges, we propose a *Unified Video Fusion framework* (**UniVF**) that explicitly incorporates multi-frame learning to exploit spatial-temporal information, thereby producing informative and temporally consistent fused videos. Specifically, UniVF adopts a Transformer-based [17] encoder-decoder architecture and employs optical flow [18] to warp features from adjacent frames to the current one, effectively capturing temporal dependencies and integrating spatio-temporal relations. A dedicated temporal consistency loss further complements the standard fusion loss based on spatial similarity to suppress flickering and promote temporal continuity across frames.

We then propose a comprehensive *Video Fusion Benchmark* (**VF-Bench**) that covers four video fusion tasks: *multi-exposure video fusion*, *multi-focus video fusion*, *infrared-visible video fusion*, and *medical video fusion*. For the first two tasks, where paired videos are difficult to acquire directly, we propose novel data generation paradigms: To create multi-exposure data, we utilize 10-bit high dynamic range (HDR) videos, convert the encoded video signals into the linear light domain via the Electro-Optical Transfer Function (EOTF), and perform exposure adjustments to generate diverse exposure pairs; For the multi-focus case, we leverage advances in video depth estimation to simulate the optical focusing process, thereby creating realistic multi-focus video pairs from standard videos. For the latter two tasks, where realistic data synthesis is infeasible, we carefully curate existing datasets by defining objective selection criteria and conducting manual screening to ensure data quality and sufficiently accurate alignment. Moreover, we develop a comprehensive suite of evaluation metrics that cover both the (per-frame) spatial quality and the (frame-to-frame) temporal consistency of a fused video, providing a more holistic evaluation protocol.

Our main contributions can be summarized as follows, with an illustrative overview in Fig. 1:

- We propose a novel *Unified Video Fusion framework*, **UniVF**, that explicitly incorporates multi-frame learning and cross-frame feature warping to exploit spatial-temporal information, producing informative and temporally consistent videos.
- We construct the first comprehensive *Video Fusion Benchmark*, **VF-Bench**, by carefully designed data generation strategies and rigorous selection from existing datasets. VF-Bench provides well-aligned, high-quality video pairs across four representative video fusion tasks (multi-exposure, multi-focus, infrared-visible, and medical video fusion).
- To train UniVF on VF-Bench, we introduce a temporal consistency loss alongside the conventional image fusion losses, to suppress flickering and ensure smooth frame transitions in fused videos.
- We establish a comprehensive evaluation protocol for video fusion, integrating both spatial quality and temporal consistency metrics for a thorough assessment.

Experiments on VF-Bench demonstrate that our UniVF achieves state-of-the-art (SOTA) video fusion performance across all four sub-tasks, setting a strong baseline for future research in video fusion.

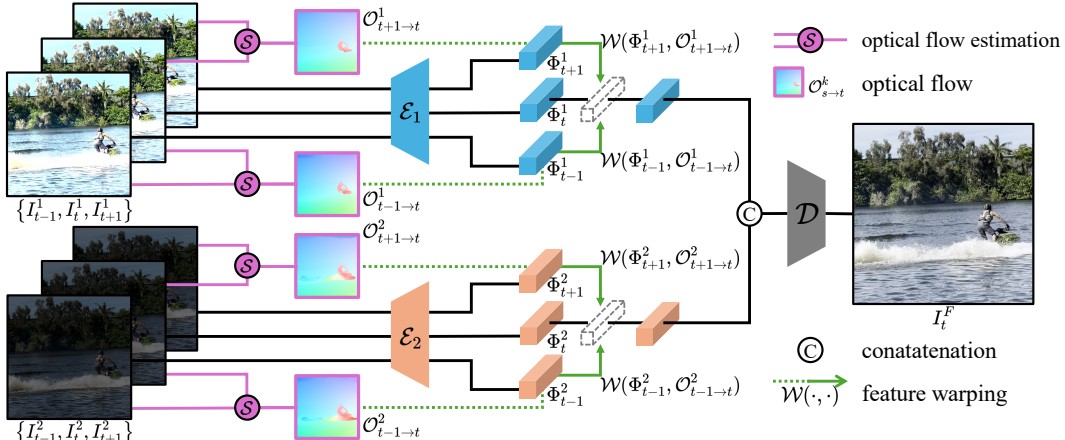

Figure 2: Detailed illustration of our UniVF architecture.

## 2 Related Work

In the era of deep learning, neural networks are frequently used in image fusion to extract source features, merge features, and reconstruct the fused image [19, 1, 4]. Image fusion algorithms are commonly categorized into two categories: discriminative [20–23] and generative [24, 3]. Discriminative models, utilizing feature extractors such as CNNs [25–30] and Transformers [31, 29], employ model-driven [20–23] or data-driven [32, 1, 33] approaches to obtain features from source images in the image domain, frequency domain, or feature space [34–36]. After information interaction and fusion within the feature space, these models ultimately learn a mapping from the source images to the fused image. On the other hand, generative fusion methods, like GANs [24, 37] and Diffusion [13, 2] models, perform modeling of the latent space manifold to minimize the distributional gap between source and fused images, providing more details in the fused results. Additionally, upstream image registration contributes to robust performance if inputs are misaligned [38–41]. Guidance from downstream tasks [42–44], such as object detection [3, 45, 46] and semantic segmentation [47–50], allows the model to learn more semantically relevant information. Furthermore, unified fusion models leverage inter-task synergy [51–55], while meta-learning can help to better adapt the loss function and the feature extractor [54, 43, 46]. Vision-language models, through their more explicit semantics, offer more flexible guidance [56, 57]. Recently, video fusion, as a further advancement of image fusion, has been demonstrated for infrared and RGB inputs [58, 59]. Moreover, multi-exposure sequences with alternating exposure times enable HDR video reconstruction [60–63]. However, a unified video fusion framework and benchmark are still lacking.

## 3 UniVF: A Unified Video Fusion Framework

**Overview**. Given a pair of video sequences $\mathcal{V}_1 = \{I_t^1\}_{t=1}^T$ and $\mathcal{V}_2 = \{I_t^2\}_{t=1}^T$, where $T$ is the total number of frames, the goal of video fusion is to generate a fused video $\mathcal{V}_F = \{I_t^F\}_{t=1}^T$ that integrates complementary information from both inputs. In the following, we introduce our *Unified Video Fusion framework*, **UniVF**, and describe how it utilizes spatial information within each frame and temporal dependencies between adjacent frames to produce temporally coherent fused videos.

### 3.1 UniVF Details

The proposed UniVF framework is made up of four key components: a feature extractor, an optical flow estimator, a feature warping module, and a feature decoder, which are responsible for extracting frame-wise features, estimating their displacements from frame to frame, aligning them, and reconstructing the fused frames, respectively. An illustration of the architecture is shown in Fig. 2.

**Feature Extraction**. The goal of this component is to extract domain-specific and spatially rich features from each source video stream. Given a pair $\{\mathcal{V}_1, \mathcal{V}_2\}$, for each time step $t$, we extract a snippet of three consecutive frames from each source: $\{I_{t-1}^k, I_t^k, I_{t+1}^k\}$ where $k \in \{1, 2\}$. Each video stream has a dedicated encoder $\mathcal{E}_k(\cdot, \cdot, \cdot)$ consisting of several Restormer blocks [17], which is shared

across the three frames of the same source:

$$\Phi_{t-1}^k, \Phi_t^k, \Phi_{t+1}^k = \mathcal{E}_k(I_{t-1}^k, I_t^k, I_{t+1}^k), \ k \in \{1, 2\}. \tag{1}$$

**Optical Flow and Feature Warping**. The difference between video fusion and single-frame image fusion lies in the ability to jointly reason over multiple consecutive frames. By exploiting information from preceding and succeeding frames, video fusion can capture dynamics and enhance feature extraction in the current frame. Thus, inspired by [64, 65], we explicitly estimate dense optical flow to align features from adjacent frames to the current time step. Specifically, given two consecutive frames $I_s^k$ and $I_t^k$ ($s \in \{t-1, t+1\}$), SEA-RAFT $\mathcal{S}(\cdot, \cdot)$ [18], a SOTA optical flow estimator, predicts the bidirectional flow $\mathcal{O}_{s \to t}^k$:

$$\mathcal{O}_{s \to t}^k = \mathcal{S}(I_s^k, I_t^k), \ k \in \{1, 2\}, \ s \in \{t-1, t+1\}. \tag{2}$$

Each optical flow field represents the motion of pixels from one frame to another, where each flow vector indicates the displacement of a pixel to its corresponding location in the neighboring frame. We choose SEA-RAFT [18] for its combination of simplicity, efficiency, and accuracy, which suits our video fusion scenario. Then, to temporally align features, UniVF performs feature warping based on these estimated flows via (differentiable) bilinear sampling. The bidirectional flow fields $\mathcal{O}_{s \to t}^k$ are used to warp the deep features from adjacent frames to the current time step:

$$\widetilde{\Phi}_{s \to t}^k = \mathcal{W}(\Phi_s^k, \mathcal{O}_{s \to t}^k), \ k \in \{1, 2\}, \ s \in \{t-1, t+1\}, \tag{3}$$

where $\mathcal{W}(\cdot, \mathcal{O})$ denotes warping according to the flow field $\mathcal{O}$. Warped features $\widetilde{\Phi}_{s \to t}^k$ are temporally aligned with the target frame and serve as motion-compensated inputs for subsequent fusion.

**Fusion and Reconstruction**. The $3 \times 2$ feature maps from both sources, warped to a common reference, are concatenated along the channel dimension and fed to the Restormer-based [17] decoder $\mathcal{D}(\cdot)$, which is tasked with modeling long-range dependencies in both the spatial and temporal dimensions:

$$\Phi_t^F = \text{Concat}\left(\Phi_t^1, \Phi_t^2, \widetilde{\Phi}_{t-1 \to t}^1, \widetilde{\Phi}_{t+1 \to t}^1, \widetilde{\Phi}_{t-1 \to t}^2, \widetilde{\Phi}_{t+1 \to t}^2\right), I_t^F = \mathcal{D}(\Phi_t^F). \tag{4}$$

Finally, the per-frame fusion results $I_t^F$ are reassembled into a fused video sequence.

## 4 VF-Bench: A Video Fusion Benchmark

**Overview**. To advance the development and evaluation of video fusion techniques and to promote further research into the topic, we have put together a *Video Fusion Benchmark*, **VF-Bench**, a comprehensive benchmarking suite that includes four different video fusion scenarios: multi-exposure, multi-focus, infrared-visible, and medical video fusion. Examples are shown in Fig. 1. The dataset offers a large collection of paired video sequences with good quality, and precisely aligned to support both model training and testing. In the following we describe our data generation pipeline and the selection criteria. For additional visualizations, as well as further details about data preparation, please refer to Appendices A and B.

**Multi-Exposure Video Fusion**. To construct multi-exposure video pairs, we propose a novel data processing pipeline with which we synthetically generate different exposure levels from 10-bit HDR source videos by adjusting exposure parameters, see Fig. 3(a). The use of 10-bit HDR videos is advantageous because it preserves a wide dynamic range, ensuring that even after exposure adjustments and potential quality degradation, details are retained that would be lost with 8-bit SDR sources. We start from the YouTube-HDR Dataset [66], a large-scale collection of short-form 10-bit HDR videos sourced from YouTube. From >2000 candidates, we manually curated 500 scenes with an average of 150 frames, choosing those with rich visual content, vivid colors, and free from watermarks or video effects. These were further divided into 450 for training and 50 for testing.

Since exposure depends approximately linearly on scene radiance, it is essential to perform exposure adjustment in the linear light domain. In this way one accurately simulates radiometric changes and avoids distortions introduced by non-linear gamma-encoding. Therefore, we first convert the encoded video signals with the Electro-Optical Transfer Function (EOTF) and transform the video into a linear color space. Exposure adjustments are then applied in this linear domain by $\pm 3$ EV (exposure value), simulating over-exposed and under-exposed conditions. To produce 8-bit videos,

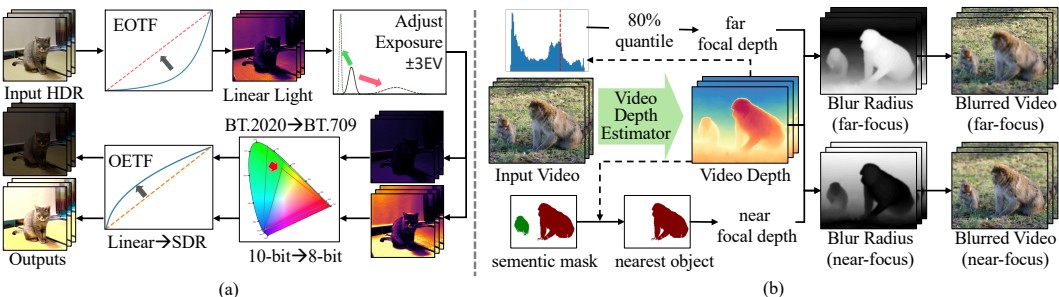

Figure 3: The proposed data generation paradigms for (a) multi-exposure video pair and (b) multi-focus video pair for our VF-Bench.

we then perform gamut mapping from the 10-bit BT.2020 color space to the 8-bit BT.709 color space. The adjusted videos are mapped to 8-bit SDR video pairs using the BT.709 standard Gamma Opto-Electronic Transfer Function (OETF) [67], to obtain paired over- and under-exposed video sequences. The described process closely resembles real-world multi-exposure video capture, where a single scene is recorded at different exposure settings, while ensuring consistent color mapping and precise spatial-temporal alignment across exposure levels. More details about this process are provided in Appendix A.1.

**Multi-Focus Video Fusion**. Existing multi-focus image fusion datasets primarily rely on light field cameras [68, 69], manually labeled focal masks [70, 71] or blur simulation based on foreground-background semantic segmentation masks [72]. The former two approaches are expensive and difficult to scale up, whereas the last one does not follow the physical process: focal planes and associated circles of confusion (CoC), by definition, depend on continuous scene depth [73] rather than on semantic labels.

Therefore, we propose to utilize depth maps to construct a multi-focus video dataset, by estimating the per-pixel blur radius, see Fig. 3(b). We use the DAVIS dataset [74] as video source for our experiments. Specifically, 150 videos are split into 120 training scenes and 30 test scenes, with an average length of 70 frames. We run a single-view video depth estimator [75] on them to obtain dense (inverse) depth. Given a focal depth, the CoC for pixel $i$ can be calculated as:

$$CoC^i = Af|(D_f - D^i)/(D_f - f)|/D^i \approx d_f|1 - d^i/d_f|\sigma, \tag{5}$$

with $A$ the aperture, $f$ the focal length, $D_f$ the focal depth, and $D^i$ the depth of scene pixel $i$. To account for unknown camera metadata, the CoC is approximated by the estimated normalized inverse depth $d^i$, the given normalized focal depth $d_f$, and a constant blur strength factor $\sigma$. Further details on the derivation of Eq. (5) can be found in Appendix A.2.

We select two normalized focal depth values, $d_f^{far}$ and $d_f^{near}$, from the first frame of each video, representing the background and foreground focus, respectively. The background focal depth is the 20th percentile of the inverse depth values. To find the foreground focus depth, we compute an average depth value for each segmented object according to the DAVIS masks [74] and select the closest of them. Finally, a Gaussian blur is applied to each pixel with the kernel size proportional to the calculated CoC, thereby generating paired multi-focus video sequences.

**Infrared-Visible Video Fusion**. Unlike the previous two tasks, paired infrared-visible video datasets can be obtained from RGBT tracking benchmarks, whereas they are difficult to realistically simulate. To construct our video fusion dataset, we start from the VTMOT [76] dataset and implement a three-stage filtering process to ensure data quality, accurate alignment, and diversity of content. First, infrared video frames are evaluated using three objective metrics: Image Entropy, Global Contrast, and Dark Area Proportion. Frames that exhibit low entropy, insufficient contrast or excessive dark regions are considered low-quality and discarded, thus removing uninformative scenes. Second, for the RGB modality, we adopt Retinex theory to decompose each frame into illumination and reflectance components [77]. RGB frames with high illumination values, indicating sufficient ambient lighting and limited need for infrared data, are excluded, thus retaining only video pairs where the infrared and visible channels provide complementary information. Finally, we perform frame-wise fusion of the remaining video pairs using SOTA image fusion algorithms like CDDFuse [1] and EMMA [78], followed by manual inspection to validate the alignment and eliminate pairs affected by mis-registration or ghosting artifacts. Further details of the complete selection criteria can be found

in Appendix A.3. Through this process, we curate a total of 90 video scenes with, on average, 300 frames. These are randomly split into 75 training scenes and 15 testing scenes.

**Medical Video Fusion**. For medical video fusion, we rely on the Harvard Medical dataset [79] as a source, treating consecutive slices of MRI and corresponding CT, PET or SPECT volumes as video sequences. A filtering strategy similar to the one used for infrared-visible fusion is adopted to ensure data quality. Specifically, frames with large invalid regions or poor visual quality are removed, preserving only meaningful, interpretable sequences with rich visual details. As a result, we curate a total of 57 scenes with 27 frames on average, which are divided into 49 for training and 8 for testing.

## 5 Experiments

We now evaluate our UniVF on VF-Bench for all four fusion scenarios: multi-exposure fusion (MEF), multi-focus fusion (MFF), infrared-visible fusion (IVF), and medical video fusion (MVF). We first describe the experimental setup, with a particular focus on the newly proposed temporal consistency term in the training loss, as well as the temporal consistency evaluation metrics that we add to the single-frame evaluation protocol. Then, we discuss the results, which highlight that our approach already constitutes a strong baseline. Finally, we conduct ablation studies to validate our design choices. Further experimental results are shown in Appendix D due to space limitations.

### 5.1 Setup

**Loss Function**. To jointly optimize spatial fidelity and temporal consistency, we adopt a compound training loss with three terms:

$$\mathcal{L} = \mathcal{L}_{\text{spatial}} + \alpha_1 \mathcal{L}_{\text{grad}} + \alpha_2 \mathcal{L}_{\text{temp}}, \tag{6}$$

where $\{\alpha_1, \alpha_2\}$ are weight parameters, set to $\{10, 2\}$, $\{1, 0.5\}$, $\{5, 2\}$ and $\{1, 1\}$ for MEF, MFF, IVF and MVF tasks respectively, such that the terms have comparable magnitudes. The three losses are *(i) Spatial similarity:* the spatial loss $\mathcal{L}_{\text{spatial}}$ measures per-pixel reconstruction error. Specifically, for IVF and MVF tasks, following [1], $\mathcal{L}_{\text{spatial}} = \frac{1}{HW}\|I_t^F - \max(I_t^1, I_t^2)\|_1$. For MEF tasks, following [56], we set $\mathcal{L}_{\text{spatial}} = \mathcal{L}_{int} + \mathcal{L}_{\text{MEF-SSIM}}$, where $\mathcal{L}_{int} = \frac{1}{HW}\|I_t^F - mean(I_t^1, I_t^2)\|_1$, and $\mathcal{L}_{\text{MEF-SSIM}}$ borrows from [80]. For MFF task, we set $\mathcal{L}_{\text{spatial}} = \mathcal{L}_{int} = \frac{1}{HW}\|I_t^F - mean(I_t^1, I_t^2)\|_1$. *(ii) Gradient preservation:* to preserve image structures and edges, following [56], we introduce a dedicated gradient loss $\mathcal{L}_{\text{grad}} = \frac{1}{HW}\| |\nabla I_t^F| - \max(|\nabla I_t^1|, |\nabla I_t^2|)\|_1$. $\nabla$ denotes the Sobel gradient operator. *(iii) Temporal consistency:* To suppress flickering and ensure smooth transitions across frames, we introduce a temporal consistency loss that explicitly enforces frame-to-frame consistency, by penalizing misalignments between adjacent frames:

$$\mathcal{L}_{\text{temp}} = \mathbb{E}_{p \in M_{\text{prev}}^t}\left[\left|I_t^F(p) - \mathcal{W}\left(I_{t-1}^F, \mathcal{O}_{t-1 \to t}^F\right)(p)\right|_1\right]$$
$$+ \mathbb{E}_{p \in M_{\text{next}}^t}\left[\left|I_t^F(p) - \mathcal{W}\left(I_{t+1}^F, \mathcal{O}_{t+1 \to t}^F\right)(p)\right|_1\right], \tag{7}$$

where $I_t^F$ is the fused video sequence from Eq. (4), and $\mathcal{W}(\cdot, \mathcal{O})$ denotes warping with the optical flow field $\mathcal{O}$. $M_{\text{prev}}^t$ and $M_{\text{next}}^t$ are validity masks at time step $t$ that indicate regions with reliable flow estimates, as detailed below. This loss term implements the strong temporal continuity of videos with reasonable frame rates, by enforcing consistency between the current fused frame and its two warped neighbors, so as to reduce flickering and abrupt changes.

**Validity masks**. To improve the robustness of the temporal consistency loss term $\mathcal{L}_{\text{temp}}$ in Eq. (7) and avoid unreliable gradients, we introduce validity masks $\{M_{\text{prev}}^t, M_{\text{next}}^t\}$ to identify well-aligned, non-occluded regions between adjacent frames. Each mask is derived via a forward–backward flow consistency check: given the forward flow $\mathcal{O}_{t \to t+1}$ and backward flow $\mathcal{O}_{t+1 \to t}$, the latter is first warped onto the coordinate space of frame $t$ as $\widehat{\mathcal{O}}_{t+1 \to t}(p) = \mathcal{W}(\mathcal{O}_{t+1 \to t}, \mathcal{O}_{t \to t+1}(p))$. Since the backward flow is defined in the pixel space of frame $t+1$ while the forward flow originates from frame $t$, warping enables both flows to be compared in the same coordinate space. The consistency error is computed as $\Delta(p) = \left\|\mathcal{O}_{t \to t+1}(p) + \widehat{\mathcal{O}}_{t+1 \to t}(p)\right\|_2$, and the binary mask is defined as $M(p) = 1$ if $\Delta(p) < \epsilon$, otherwise 0. $\epsilon$ is a predefined threshold (set to 1.0 in our implementation). Intuitively, this verifies whether a pixel can move forward and then return along the estimated flow paths with minimal deviation. Large inconsistencies typically indicate occlusions, motion boundaries, or flow

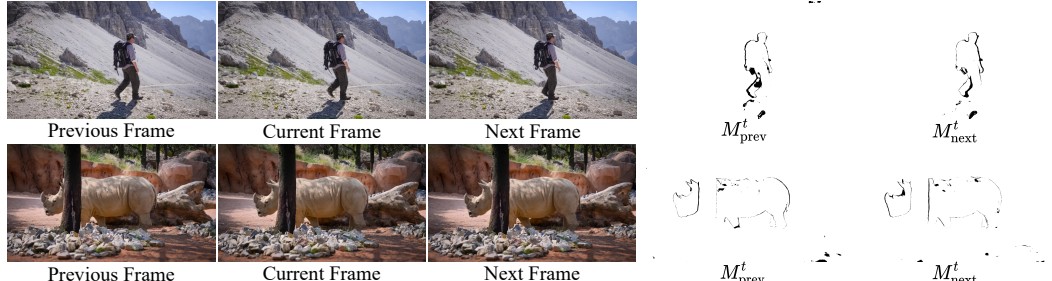

Figure 4: Previous, current, and next frames with their corresponding validity masks $M_{\text{prev}}^t$ and $M_{\text{next}}^t$. Black regions denote invalid or unreliable areas, corresponding to poorly aligned or occluded pixels that are excluded from the temporal consistency computation.

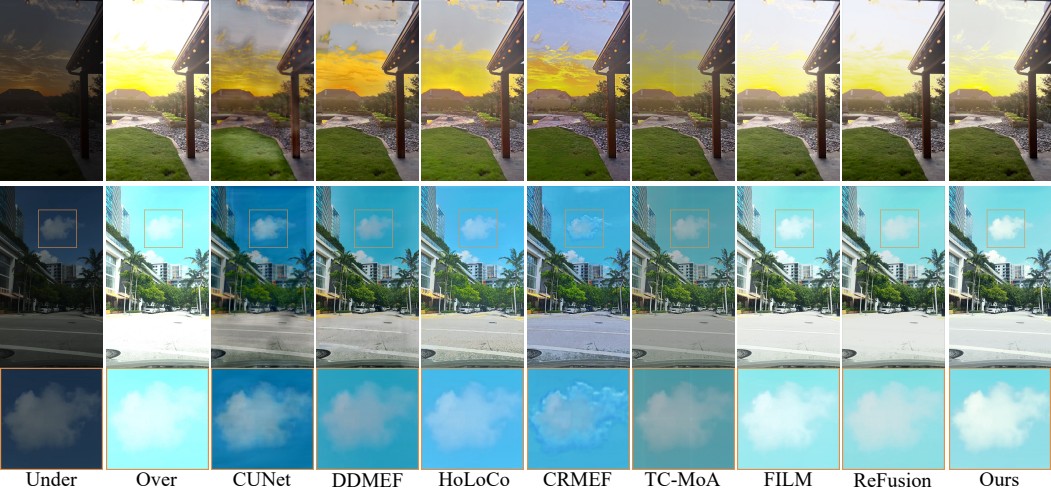

| Under | Over | CUNet | DDMEF | HoLoCo | CRMEF | TC-MoA | FILM | ReFusion | Ours |

Figure 5: Qualitative comparison of fusion outcomes for multi-exposure video fusion.

ambiguities. By restricting the temporal consistency loss to these valid regions, UniVF effectively reduces flickering and enforces smooth temporal transitions. Visual examples of $\{M_{\text{prev}}^t, M_{\text{next}}^t\}$ are shown in Fig. 4.

**Training details**. We ran our experiments on a machine equipped with a single NVIDIA GeForce RTX 4090 GPU. The loss is minimized with Adam, starting with a learning rate of $10^{-4}$ that decays exponentially to 1% of its initial value over the course of 20k iterations. Training uses a batch size of 32, with gradient accumulation. As our network architecture, we adopt Restormer blocks [17] in both the encoder $\mathcal{E}_k(\cdot)$ and decoder $\mathcal{D}(\cdot)$ components. Each block contains 8 attention heads and has a feature dimension of 32. Both the encoders and decoder are configured with 4 stacked blocks.

**Metrics**. *(i) Spatial domain evaluation metrics:* We adopt four widely used quantitative metrics: VIF (Visual Information Fidelity), SSIM (Structural Similarity Index), MI (Mutual Information), and $Q^{AB/F}$. These indicators measure perceptual fidelity, structural similarity, mutual information content, and edge preservation, respectively. In all cases, a higher value means better fusion of the complementary information from the two sources. For further details, see [81]. *(ii) Temporal domain consistency metrics:* To assess the temporal consistency and motion smoothness of fused videos, we proposed two complementary evaluation metrics.

**Bi-Directional Self-Warping Error (BiSWE)**: As a reference-free metric, BiSWE is designed to quantify frame-to-frame *temporal alignment errors* within a video sequence. Given a video clip $\{I_{t-1}, I_t, I_{t+1}\}$, we compute the optical flow fields $\mathcal{O}_{s \to t} = \mathcal{S}(I_s, I_t)$, $s \in \{t-1, t+1\}$ with SEA-RAFT $\mathcal{S}(\cdot, \cdot)$ [18]. Validity masks $\{M_{\text{prev}}^t, M_{\text{next}}^t\}$ are applied to exclude unreliable regions based on forward-backward consistency. The BiSWE value is computed as

$$\text{BiSWE} = \mathbb{E}_{p \in M_{\text{prev}}^t}[|I_t(p) - \mathcal{W}(I_{t-1}, \mathcal{O}_{t-1 \to t})(p)|_1]$$
$$+ \mathbb{E}_{p \in M_{\text{next}}^t}[|I_t(p) - \mathcal{W}(I_{t+1}, \mathcal{O}_{t+1 \to t})(p)|_1], \tag{8}$$

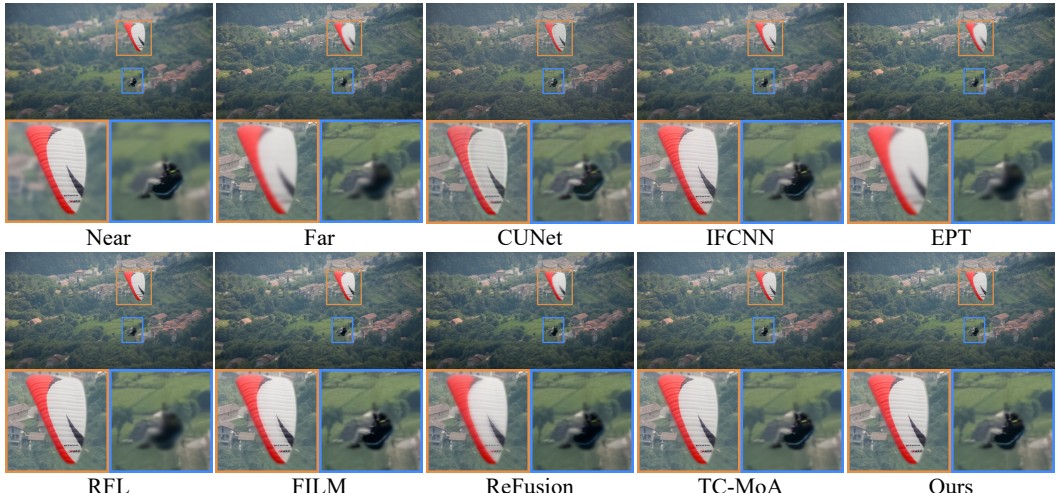

Figure 6: Qualitative comparison of fusion outcomes for multi-focus video fusion.

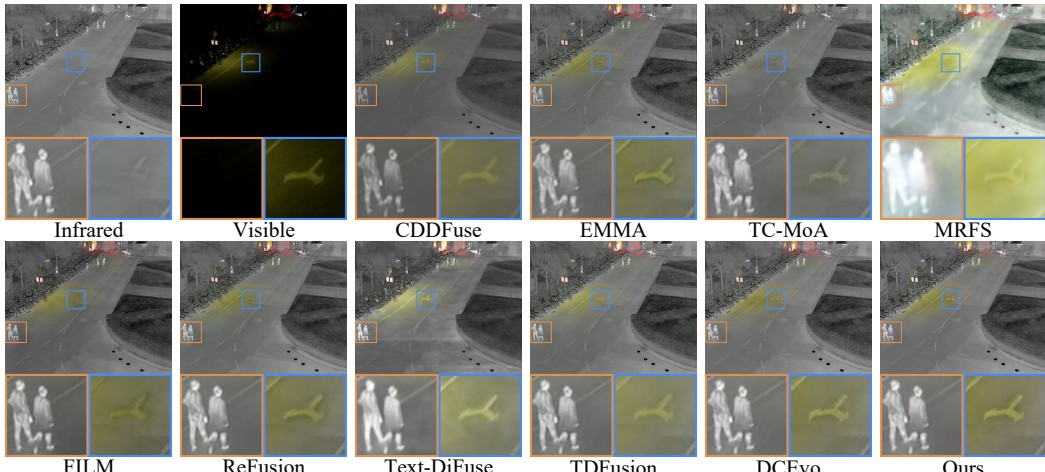

Figure 7: Qualitative comparison of fusion outcomes for infrared-visible video fusion.

where $\mathcal{W}(\cdot, \mathcal{O})$ is the warping function guided by the flow $\mathcal{O}$, and $\mathbb{E}$ denotes averaging over valid pixels. Lower BiSWE indicates improved temporal alignment.

**Motion Smoothness with Dual Reference Videos (MS2R)**: To assess the naturalness and consistency of motion transitions, MS2R evaluates the coherence of *flow changes* in the fused video and two reference sequences. The flow change is defined as the difference between consecutive flows within a clip. Intuitively, it compares the accelerations of objects projected onto the image plane. The MS2R score is defined as

$$\text{MS2R} = \mathbb{E}_p \left[ \left| \Delta \mathcal{O}^F(p) - \Delta \mathcal{O}^{R_1}(p) \right|_1 \right] + \mathbb{E}_p \left[ \left| \Delta \mathcal{O}^F(p) - \Delta \mathcal{O}^{R_2}(p) \right|_1 \right], \tag{9}$$

where $\Delta \mathcal{O}^F = \mathcal{O}_{1 \to 2}^F - \mathcal{O}_{0 \to 1}^F$. $\mathcal{O}_{1 \to 2}^F$ and $\mathcal{O}_{0 \to 1}^F$ are also obtained from $\mathcal{S}(\cdot, \cdot)$ [18]. $\Delta \mathcal{O}^{R_1}, \Delta \mathcal{O}^{R_2}$ are computed similarly from the two reference sequences. A lower MS2R indicates smoother and more natural motion trajectories in the fusion video.

## 5.2 Video Fusion Experiments

**Multi-Exposure Video Fusion**. We ran experiments on the MEF branch of VF-Bench. Tested methods include CUNet [9], DDMEF [82], HoLoCo [83], CRMEF [84] , TC-MoA [55], FILM [56] and ReFusion [54]. As illustrated in Tab. 1 and Fig. 5, UniVF consistently achieves superior

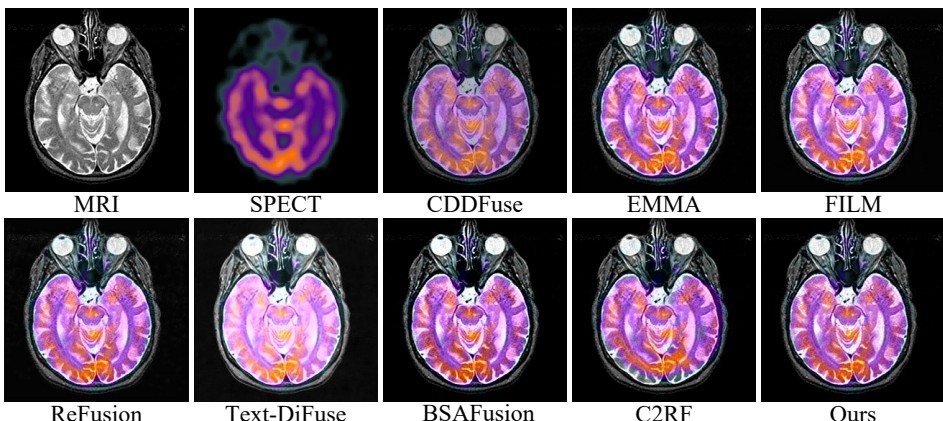

| MRI | SPECT | CDDFuse | EMMA | FILM |
| --- | --- | --- | --- | --- |
| ReFusion | Text-DiFuse | BSAFusion | C2RF | Ours |

Figure 8: Qualitative comparison of fusion outcomes for medical video fusion.

Table 1: Quantitative evaluation results for the MEF and MFF task. The red and blue highlights indicate the highest and second-highest scores.

| | VF-Bench Multi-Exposure Video Fusion Branch | | | | | | | VF-Bench Multi-Focus Video Fusion Branch | | | | | |
| --- | --- | --- | --- | --- | --- | --- | --- | --- | --- | --- | --- | --- | --- |
| | VIF↑ | SSIM↑ | MI↑ | Qabf↑ | BiSWE↓ | MS2R↓ | | VIF↑ | SSIM↑ | MI↑ | Qabf↑ | BiSWE↓ | MS2R↓ |
| CUNet | 0.58 | 0.67 | 2.26 | 0.39 | 7.12 | 0.42 | CUNet | 0.56 | 0.88 | 3.74 | 0.53 | 6.79 | 1.47 |
| DDMEF | 0.72 | 0.94 | 2.71 | 0.65 | 10.06 | 1.04 | IFCNN | 0.69 | 0.89 | 4.93 | 0.70 | 6.25 | 1.38 |
| HoLoCo | 0.39 | 0.79 | 2.30 | 0.25 | 10.52 | 0.55 | RFL | 0.78 | 0.90 | 6.29 | 0.78 | 5.96 | 1.11 |
| CRMEF | 0.64 | 0.95 | 2.61 | 0.61 | 8.68 | 0.42 | EPT | 0.77 | 0.90 | 6.31 | 0.78 | 5.97 | 1.14 |
| TC-MoA | 0.76 | 0.98 | 2.94 | 0.71 | 7.78 | 0.34 | TC-MoA | 0.77 | 0.90 | 5.46 | 0.76 | 5.99 | 1.13 |
| FILM | 0.78 | 0.98 | 4.39 | 0.71 | 8.27 | 0.34 | FILM | 0.76 | 0.89 | 5.02 | 0.75 | 6.32 | 1.28 |
| ReFus | 0.75 | 0.97 | 3.89 | 0.70 | 7.95 | 0.33 | ReFus | 0.73 | 0.90 | 4.95 | 0.73 | 5.80 | 1.28 |
| Ours | 0.82 | 0.99 | 4.45 | 0.72 | 6.40 | 0.33 | Ours | 0.79 | 0.90 | 6.32 | 0.79 | 5.95 | 1.08 |

quantitative and qualitative performance, effectively balancing dynamic range expansion, contrast enhancement, and image quality preservation across multiple exposure levels.

**Multi-Focus Video Fusion**. For the experiments on the MFF branch of VF-Bench, the tested methods include CUNet [9], IFCNN [85], RFL [86], EPT [87], TC-MoA [55], FILM [56], and ReFusion [54]. As shown in Fig. 6 and Tab. 1, our UniVF baseline again achieves the best performance both qualitatively and quantitatively, accurately identifying focused regions and producing sharp fusion results free of artifacts, across both the foreground and background.

Notably, both training and testing for the MEF and MFF branches were conducted on the 2K-resolution version of the dataset. Considering potential computational resource constraints, we also report results on a low-resolution version in the Appendix C for reference.

**Infrared-Visible Video Fusion**. We conducted experiments on the IVF branch of VF-Bench, with CDDFuse [88], EMMA [14], TC-MoA [55], MRFS [50], FILM [56], ReFusion [54], Text-DiFuse [57], TDFusion [43] and DCEvo [42]. The results once more show superior performance of UniVF, in terms of both visual quality and quantitative metrics. As illustrated in Fig. 7, the method faithfully preserves critical thermal and structural details, enhances object visibility and reduces noise in low-light conditions. Quantitative comparisons in Tab. 2 further confirm that UniVF consistently has an edge in most metrics, highlighting its robustness across diverse scenes and object types.

**Medical Video Fusion**. On the MVF branch of VF-Bench, we evaluate a range of methods including CDDFuse [88], EMMA [14], FILM [56], ReFusion [54], Text-DiFuse [57], BSAFusion [89] and C2RF [38]. As can be seen in Fig. 8 and Tab. 3, UniVF once more effectively preserves fine-grained textures from MRI images, while simultaneously enhancing and maintaining the salient intensities of the CT, PET or SPECT modality. The fused results exhibit clear anatomical details and clean tissue boundaries from the MRI source, along with distinct color distributions originating from the SPECT images to support clinical diagnosis.

**Ablation Studies**. To explore the contribution of each key component within UniVF, we conducted ablation studies on the IVF task, with results summarized in Tab. 4. In Exp. I, we removed the feature warping module, *i.e.*, multi-frame features are directly concatenated along the channel dimension and fed into the decoder, without optical flow correction. In Exp. II, both the feature warping module

Table 2: Quantitative evaluation for the IVF task.

| | VIF ↑ | SSIM ↑ | MI ↑ | Qabf ↑ | BiSWE ↓ | MS2R ↓ |
|---|---|---|---|---|---|---|
| | **VF-Bench Infrared-Visible Video Fusion Branch** | | | | | |
| CDDF | 0.37 | 0.64 | 2.41 | 0.54 | 5.12 | 0.37 |
| EMMA | 0.37 | 0.63 | 2.01 | 0.58 | 4.79 | 0.37 |
| TC-MoA | 0.37 | 0.64 | 2.05 | 0.60 | 4.68 | 0.38 |
| MRFS | 0.27 | 0.55 | 1.48 | 0.34 | 6.09 | 0.38 |
| FILM | 0.40 | 0.63 | 2.05 | 0.64 | 4.78 | 0.37 |
| ReFus | 0.42 | 0.64 | 2.27 | 0.67 | 4.64 | 0.36 |
| Text-D | 0.30 | 0.60 | 1.64 | 0.39 | 10.63 | 0.40 |
| TDFusion | 0.45 | 0.64 | 2.34 | 0.67 | 4.35 | 0.36 |
| DCEvo | 0.43 | 0.64 | 2.44 | 0.66 | 4.57 | 0.37 |
| Ours | 0.44 | 0.64 | 2.47 | 0.68 | 3.94 | 0.35 |

Table 3: Quantitative evaluation for the MVF task.

| | VIF ↑ | SSIM ↑ | MI ↑ | Qabf ↑ | BiSWE ↓ | MS2R ↓ |
|---|---|---|---|---|---|---|
| | **VF-Bench Medical Video Fusion Branch** | | | | | |
| CDDF | 0.29 | 0.76 | 1.80 | 0.59 | 26.33 | 1.34 |
| EMMA | 0.29 | 0.68 | 1.73 | 0.60 | 30.00 | 1.98 |
| FILM | 0.33 | 0.36 | 1.83 | 0.67 | 32.04 | 1.59 |
| ReFus | 0.31 | 0.32 | 1.74 | 0.67 | 32.85 | 1.74 |
| Text-D | 0.24 | 0.21 | 1.58 | 0.52 | 34.09 | 1.96 |
| BSAF | 0.28 | 0.63 | 1.69 | 0.58 | 34.73 | 1.66 |
| C2RF | 0.30 | 0.73 | 1.75 | 0.59 | 32.67 | 2.06 |
| Ours | 0.35 | 0.76 | 2.00 | 0.68 | 29.61 | 1.30 |

Table 4: Ablation experiments results, with red representing the best values.

| Descriptions | Configurations | | | Metrics | | | | | |
|---|---|---|---|---|---|---|---|---|---|
| | feature warping | multi-inputs | $\mathcal{L}_{\text{temp}}$ | VIF ↑ | SSIM ↑ | MI ↑ | Qabf ↑ | BiSWE ↓ | MS2R ↓ |
| Exp. I: w/o feature warping | | ✓ | ✓ | 0.40 | 0.63 | 2.44 | 0.66 | 4.18 | 0.36 |
| Exp. II: w/o warping & multi-inputs | | | ✓ | 0.38 | 0.61 | 2.07 | 0.64 | 4.46 | 0.37 |
| Exp. III: w/o $\mathcal{L}_{\text{temp}}$ | ✓ | ✓ | | 0.42 | 0.65 | 2.38 | 0.65 | 5.79 | 0.39 |
| UniVF (Ours) | ✓ | ✓ | ✓ | 0.44 | 0.64 | 2.47 | 0.68 | 3.94 | 0.35 |

and multi-frame inputs were removed, reverting the model to a conventional frame-by-frame fusion scheme. To ensure a fair comparison, we increased the number of Restormer blocks to maintain an equivalent total parameter count. In Exp. III, while retaining the original multi-frame fusion structure with feature warping, we switched off the temporal consistency loss $\mathcal{L}_{\text{temp}}$ during training.

Taken together, the ablation experiments demonstrate the necessity of each component in our scheme. Specifically, multi-frame feature warping enhances temporal coherence and overall fusion quality, while the temporal consistency loss further ensures smooth transitions across consecutive frames. Their combination yields superior video fusion compared to simplified or frame-wise baselines.

## 6  Conclusion

We have presented **UniVF**, a unified framework for video fusion that explicitly leverages multi-frame learning and optical flow-based feature warping to exploit both spatial and temporal information. To ensure temporal coherence of the fused results, we introduce a custom temporal consistency loss that suppresses flickering and enforces smooth frame transitions. Additionally, we have introduced a comprehensive **VF-Bench**, to our knowledge the first benchmark for video fusion. It covers four representative tasks with carefully constructed, paired video datasets and a holistic evaluation protocol, including dedicated temporal consistency metrics. Extensive experiments demonstrate that UniVF, with its straightforward design, achieves SOTA performance across all four tasks. We hope that our benchmark, together with the strong baseline of our fusion framework, encourages further research into video fusion and lays a solid foundation for it.

## Acknowledgments

This work was supported by Huawei Technologies Oy (Finland), and by a SwissAI Compute Grant.

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

# A Further Details of Data Preparation in VF-Bench

## A.1 Explanation of Key Terms in Multi-Exposure Video Fusion

- **BT.709**: A widely adopted International Telecommunication Union Radiocommunication Sector (ITU-R) standard for high-definition television (HDTV) video, specifying parameters such as color primaries, transfer characteristics (OETF), and color space for 8-bit SDR (Standard Dynamic Range) video.

- **BT.2020**: An ITU-R recommendation defining parameters for ultra-high-definition television (UHDTV), including a wider color gamut, higher bit-depth (typically 10-bit or 12-bit), and enhanced color reproduction capabilities compared to BT.709. It is used for HDR (High Dynamic Range) video content.

- **Electro-Optical Transfer Function (EOTF)**: A mathematical function that defines how digital signal values are converted into visible light by a display. It transforms non-linear, gamma-encoded video signals into a linear light domain, accurately representing scene radiance. BT.2020 (HLG encoding format[1]) EOTF is defined as:

$$L = \begin{cases} \frac{V^2}{3} & 0 \leq V \leq 0.5 \\ \frac{\exp\left(\frac{V-0.5599}{0.1788}\right)+0.2847}{12} & 0.5 < V \leq 1 \end{cases} \tag{10}$$

where $V$ is the normalized video signal value and $L$ is the corresponding linear luminance.

- **Opto-Electronic Transfer Function (OETF)**: The inverse of EOTF, this function defines how light captured by a camera sensor is converted into digital video signal values. It typically applies a gamma curve to map linear scene radiance into a non-linear encoding space suitable for storage and broadcast. The OETF for BT.709 is specified as:

$$V = \begin{cases} 4.5L & 0 \leq L < 0.018 \\ 1.099L^{0.45} - 0.099 & 0.018 \leq L \leq 1 \end{cases} \tag{11}$$

where $L$ is the normalized linear light level and $V$ is the video signal value.

- **Linear Color Space**: A color space where the numerical values of pixel intensities are directly proportional to the physical light intensity in the real world. In this domain, exposure adjustments and radiometric operations can be performed accurately, as opposed to gamma-encoded, non-linear spaces where such operations would introduce distortions.

## A.2 Circle of Confusion Derivation and Approximation (Eq. (5)) in Multi-Focus Video Fusion

The blur level in optical imaging systems is characterized by the size of the Circle of Confusion (CoC), which determines the degree of defocus blur for each pixel. A larger CoC corresponds to stronger blur. Based on the thin lens equation:

$$\frac{1}{v} + \frac{1}{D} = \frac{1}{f}, \tag{12}$$

where $v$ is the image distance, $D$ is the object distance, and $f$ is the focal length of the lens. The image distance for an object at depth $D^i$ is given by:

$$v^i = \frac{fD^i}{D^i - f}. \tag{13}$$

Assuming a focus distance $D_f$, the corresponding image distance is:

$$v_f = \frac{fD_f}{D_f - f}. \tag{14}$$

As illustrated in Fig. 9, the CoC at pixel $i$ can then be computed as:

$$\text{CoC}^i = A \left| \frac{v^i - v_f}{v^i} \right|, \tag{15}$$

---

[1]Hybrid Log-Gamma (HLG) is a widely used high dynamic range (HDR) encoding format, employed in the YouTube-HDR dataset [66].

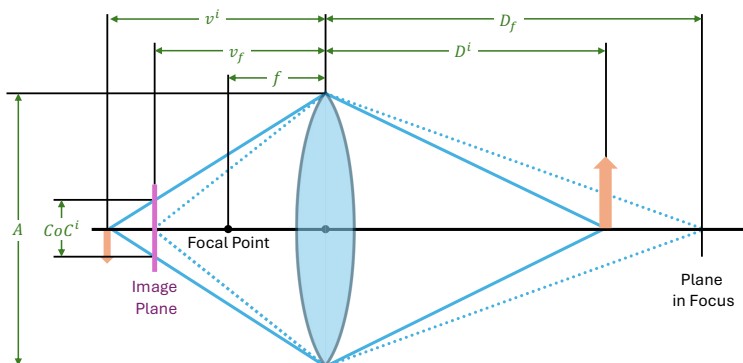

Figure 9: Illustration of the optical geometry for Circle of Confusion.

where $A$ is the aperture diameter.

Substituting the expressions for $v^i$ and $v_f$, and simplifying, we obtain:

$$\text{CoC}^i = Af \left| \frac{D^i - D_f}{D^i(D_f - f)} \right|. \tag{16}$$

To facilitate practical implementation without requiring precise camera metadata, we approximate the CoC based on estimated normalized inverse depth. Let $d^i = 1/D^i$ and $d_f = 1/D_f$. Assuming $f \ll D^i, D_f$ (a valid assumption in most video capture scenarios), and noting that the focus distance $D_f$ remains fixed within a specific video, we approximate Eq. (16) as:

$$\text{CoC}^i \propto \left| D^i - D_f \right| / D^i = \left| \frac{1}{d^i} - \frac{1}{d_f} \right| \cdot d^i. \tag{17}$$

Further simplifying yields:

$$\text{CoC}^i \propto \left| 1 - \frac{d^i}{d_f} \right|. \tag{18}$$

By introducing a global blur strength factor $\sigma$ to account for unknown camera parameters, the CoC for pixel $i$ is approximated by:

$$\text{CoC}^i \approx d_f \left| 1 - \frac{d^i}{d_f} \right| \sigma. \tag{19}$$

This approximation is justified as it preserves the relative CoC values across pixels, which is critical for simulating realistic defocus blur patterns in the absence of explicit optical parameters. Furthermore, since inverse depth typically correlates linearly with perceived defocus in monocular video sequences, this approximation remains perceptually valid. The scaling factor $\sigma$ absorbs the unknown optical constants and ensures consistent blur strength across frames. We take $\sigma = 0.025$, which is approximated by using a common camera setup.

Finally, for a frame with the longer edge length $l$ pixels, we calculate the Gaussian blur kernel size kernel$^i$ for each pixel from the calculated CoC values by:

$$\text{kernel}^i = CoC^i \cdot l. \tag{20}$$

## A.3 Selection Criteria for Infrared-Visible Video Fusion

### A.3.1 Objective Assessment for Infrared Frames

To ensure the quality of infrared-visible video fusion, an objective assessment is performed on infrared frames prior to fusion. Three quantitative metrics — *Image Entropy*, *Global Contrast*, and *Dark Area Proportion* — are used to evaluate each frame. Frames that do not meet predefined thresholds are

discarded to exclude low-quality or uninformative content. In the following, we present more details of the metrics, computation methods, and thresholds.

**Image Entropy**. Image Entropy quantifies the information content and textural complexity of a grayscale image. Higher entropy indicates richer pixel intensity distribution, while lower values indicate less informative content. It is computed as:

$$H = -\sum_{i=0}^{255} p(i) \log_2 p(i), \tag{21}$$

where $p(i)$ is the normalized histogram value of intensity $i$. The entropy of each infrared frame is calculated from its normalized histogram via Eq. (21). Frames with $H > 6$ are retained while those below this threshold are excluded due to insufficient information.

**Global Contrast**. Global Contrast is assessed by the standard deviation of pixel intensities, reflecting the overall contrast distribution in the image. A higher standard deviation indicates stronger contrast and clearer object boundaries, which is essential to highlight thermal patterns. It is computed as:

$$\sigma = \sqrt{\frac{1}{N} \sum_{i=1}^{N} (x_i - \mu)^2}, \tag{22}$$

where $x_i$ is the intensity of pixel $i$, $\mu$ the mean intensity, and $N$ the total number of pixels. Each frame's contrast is evaluated from its grayscale intensities. Frames with $\sigma > 30$ are retained; those below are discarded due to insufficient structural and thermal contrast.

**Dark Area Proportion**. Frames containing excessive dark regions often reflect poor capture conditions or insufficient thermal signals. To quantify this, we compute the *Dark Area Proportion $D$*, which represents the proportion of pixels whose intensity falls below a predefined threshold $T = 10$:

$$D = \frac{1}{N} \sum_{i=1}^{N} \mathbb{I}(I_i \leq T), \tag{23}$$

where $N$ is the total number of pixels in the frame and $\mathbb{I}(\cdot)$ is the indicator function:

$$\mathbb{I}(I_i \leq T) = \begin{cases} 1, & \text{if } I_i \leq T \\ 0, & \text{otherwise} \end{cases} \tag{24}$$

Frames with a dark ratio under 5% are retained while those exceeding this value are discarded for lacking sufficient thermal content and visibility.

**Overall Quality Scoring**. After filtering based on each of the three individual metrics, we further perform a comprehensive selection of scenes using a weighted normalized score that combines the three metrics:

$$\text{Score} = w_1 \cdot \frac{H}{H_{\max}} + w_2 \cdot \frac{\sigma}{\sigma_{\max}} + w_3 \cdot (1 - D). \tag{25}$$

$H_{\max}$ and $\sigma_{\max}$ denote the maximum values of $H$ and $\sigma$ over the entire dataset, respectively. Weights $w_1$, $w_2$, and $w_3$ satisfying $w_1 + w_2 + w_3 = 1$. Equal weights ($w_1 = w_2 = w_3 = 1/3$) are used here. This composite score offers an intuitive measure for evaluating overall frame quality. Based on this score, we discard the bottom 10% of scenes.

### A.3.2 Exclusion of High-Illumination RGB Frames

Retinex theory decouples a natural image ($I$) into an illumination component ($L$) and a reflectance component ($R$), mathematically expressed as:

$$I = L * R. \tag{26}$$

Here, $L$ represents the intensity of illumination incident upon the scene, which varies with lighting conditions. Conversely, $R$ signifies the intrinsic reflectance properties of the scene, which remain invariant to changes in illumination. This Retinex decomposition allows for the effective extraction of the scene's illumination information, facilitating subsequent filtering processes. In our work, we leverage the model proposed in [77] to extract the scene's illumination component. Following this, frames with illumination levels ranked in the top 25% based on mean($L$) values are excluded.

Table 5: Quantitative evaluation results for the low-resolution MEF (540p) and MFF (480p) task. The red and blue highlights indicate the highest and second-highest scores.

| | VF-Bench Multi-Exposure Fusion Branch (540p) | | | | | | | VF-Bench Multi-Focus Fusion Branch (480p) | | | | | |
|---|---|---|---|---|---|---|---|---|---|---|---|---|---|
| | VIF↑ | SSIM↑ | MI↑ | Qabf↑ | BiSWE↓ | MS2R↓ | | VIF↑ | SSIM↑ | MI↑ | Qabf↑ | BiSWE↓ | MS2R↓ |
| CUNet | 0.50 | 0.85 | 1.85 | 0.39 | 7.55 | 0.20 | CUNet | 0.53 | 0.86 | 3.52 | 0.68 | 10.23 | 0.42 |
| DDMEF | 0.71 | 0.95 | 2.96 | 0.66 | 8.99 | 0.71 | IFCNN | 0.68 | 0.87 | 4.85 | 0.73 | 9.37 | 0.38 |
| HoLoCo | 0.50 | 0.86 | 2.56 | 0.42 | 8.22 | 0.19 | RFL | 0.77 | 0.90 | 6.31 | 0.78 | 8.46 | 0.28 |
| CRMEF | 0.62 | 0.94 | 2.60 | 0.63 | 8.72 | 0.19 | EPT | 0.76 | 0.90 | 6.33 | 0.78 | 8.50 | 0.29 |
| TC-MoA | 0.74 | 0.99 | 2.93 | 0.72 | 7.82 | 0.16 | TC-MoA | 0.75 | 0.90 | 5.27 | 0.77 | 8.39 | 0.28 |
| FILM | 0.77 | 0.99 | 4.35 | 0.72 | 8.28 | 0.17 | FILM | 0.75 | 0.89 | 5.06 | 0.78 | 8.61 | 0.33 |
| ReFus | 0.74 | 0.97 | 3.81 | 0.72 | 7.63 | 0.16 | ReFus | 0.73 | 0.90 | 4.93 | 0.77 | 8.00 | 0.32 |
| Ours | 0.79 | 0.99 | 4.38 | 0.73 | 6.96 | 0.16 | Ours | 0.77 | 0.90 | 6.34 | 0.79 | 8.29 | 0.27 |

# B   Visualizations of the Four Branches in VF-Bench

We present visualizations of some part of the video pairs from the four branches in VF-Bench as follows:

- Dataset visualizations for *Multi-exposure Video Fusion* branch in VF-Bench are shown in Fig. 10.
- Dataset visualizations for *Multi-focus Video Fusion* branch in VF-Bench are shown in Fig. 11.
- Dataset visualizations for *Infrared-Visible Video Fusion* branch in VF-Bench are shown in Fig. 12.
- Dataset visualizations for *Medical Video Fusion* branch in VF-Bench are shown in Fig. 13.

Our VF-Bench provides high-quality data in diverse scenes, serving as a strong benchmark for future video fusion tasks.

# C   Quantitative Results on Low-Resolution MEF and MFF Branches

Considering that inference on full-resolution videos may not be suitable for small devices or scenarios with limited computational resources, we additionally conduct experiments on low-resolution versions of the MEF (540p) and MFF (480p) datasets in Tab. 5. While the primary training and evaluation of both branches are performed on the 2K-resolution datasets in the main paper, the low-resolution results presented here serve as complementary evidence to assess the consistency of performance across different input scales. The supplementary results further demonstrate that our model can consistently produce high-quality fused videos.

# D   Additional Qualitative Fusion Comparison Results

We present more fusion visualization results in the figures below and on the project homepage videos:

- More qualitative comparisons for *Multi-exposure Video Fusion* results are shown in Fig. 14.
- More qualitative comparisons for *Multi-focus Video Fusion* results are shown in Fig. 15.
- More qualitative comparisons for *Infrared-Visible Video Fusion* results are shown in Fig. 16.
- More qualitative comparisons for *Medical Video Fusion* results are shown in Fig. 17.

The visual comparisons support our observation and conclusions: our UniVF method effectively preserves fine details and texture from the source images, while comprehensively integrating information from different settings or modalities to produce richly informative fused images. The videos further show that our results exhibit superior temporal consistency, with significantly less flickering and motion incoherence.

# E   Limitations

We assume well alignment between modalities in the input videos, and we have accordingly filtered the videos in VF-Bench. However, there are, although rarely, still a few frames (1 out of 300 frames) that are not aligned. With this assumption and trained on our well-filtered data, the model may

produce artifacts or ghosting in the fused video when encountering misaligned frames, as shown in Fig. 18. In future work, we plan to incorporate alignment modules into our UniVF to improve the robustness of the fusion process against misaligned input frames while maintaining the model's efficiency and fusion quality.

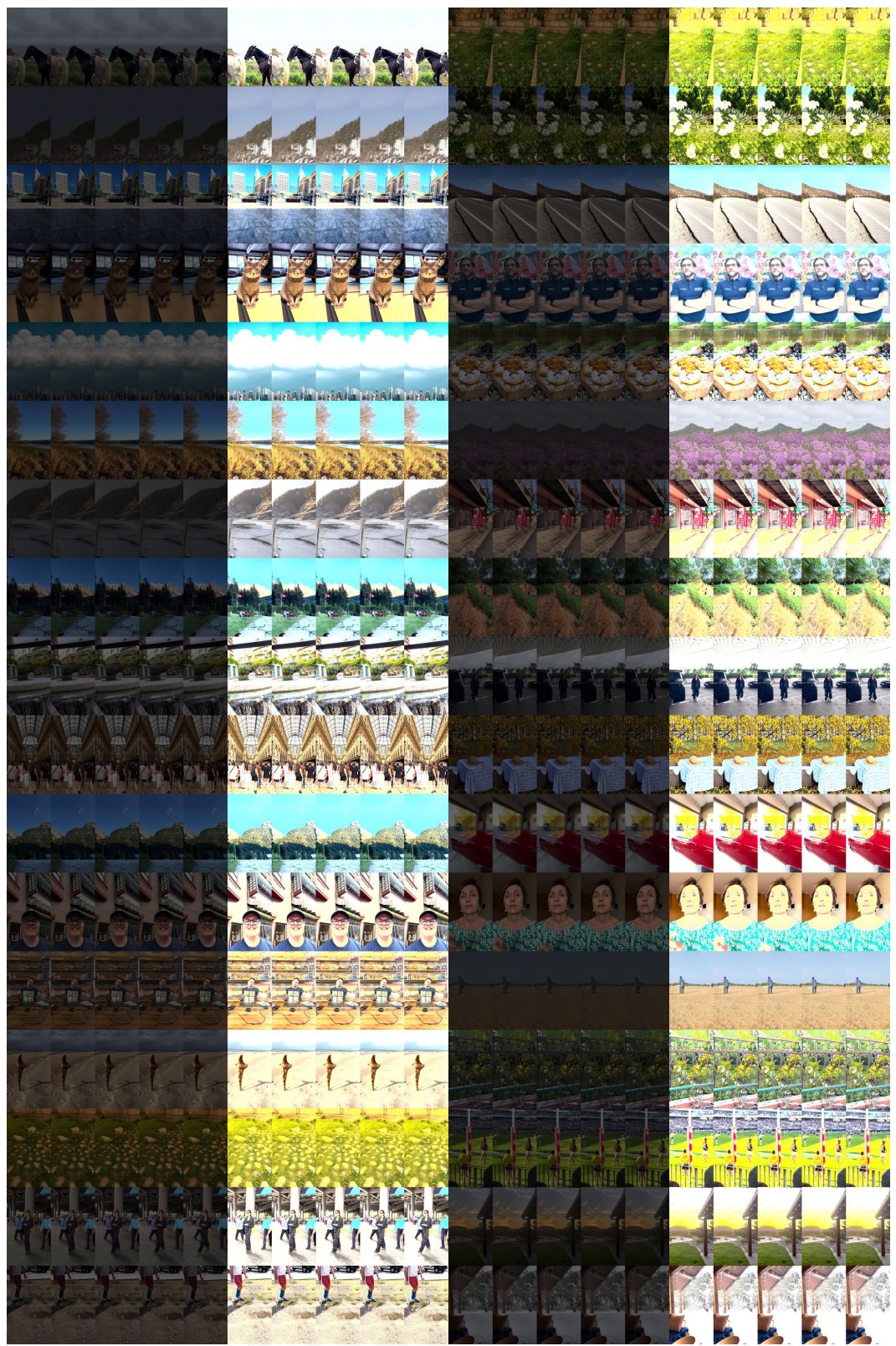

Figure 10: Dataset visualizations for *Multi-exposure Video Fusion* branch in VF-Bench. Columns 1–5 and 11–15 correspond to under-exposed video sequences, while columns 6–10 and 16–20 correspond to their respective over-exposed video sequences.

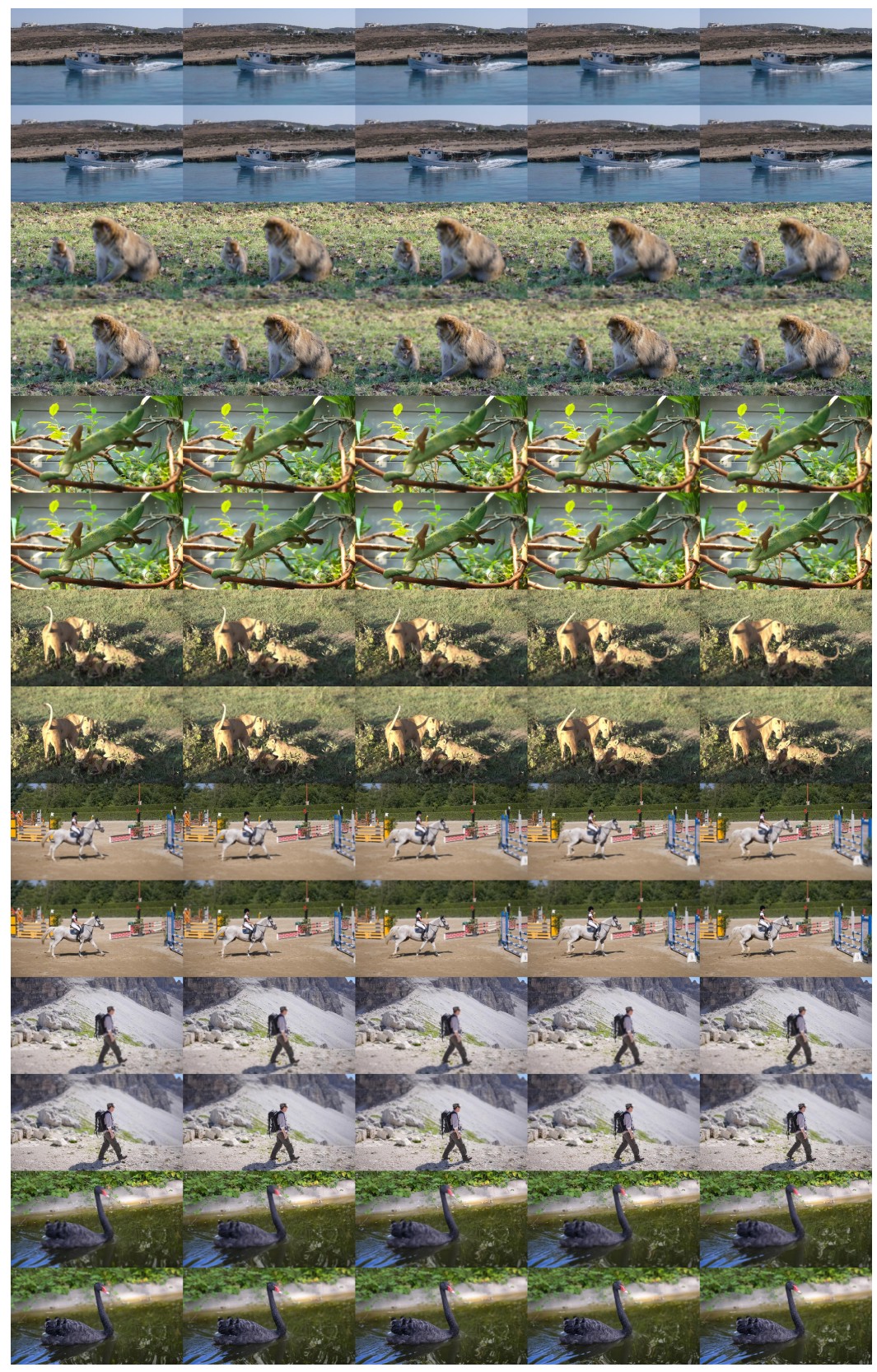

Figure 11: Dataset visualizations for *Multi-focus Video Fusion* branch in VF-Bench. Odd rows correspond to the far-focus video sequences and even rows correspond to the respective near-focus video sequences.

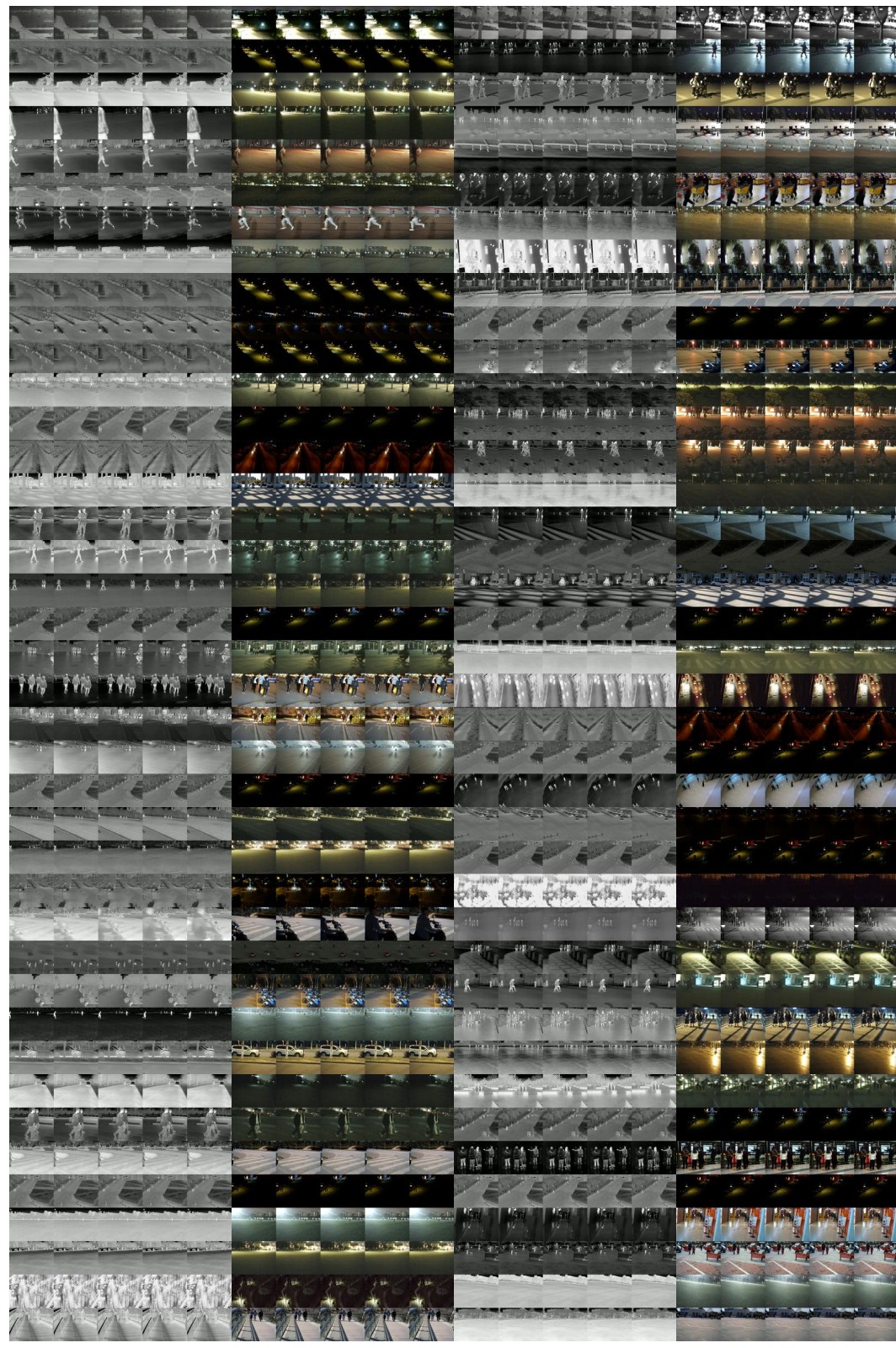

Figure 12: Dataset visualizations for *Infrared-Visible Video Fusion* branch in VF-Bench. Columns 1–5 and 11–15 correspond to infrared video sequences, while columns 6–10 and 16–20 correspond to their respective visible video sequences.

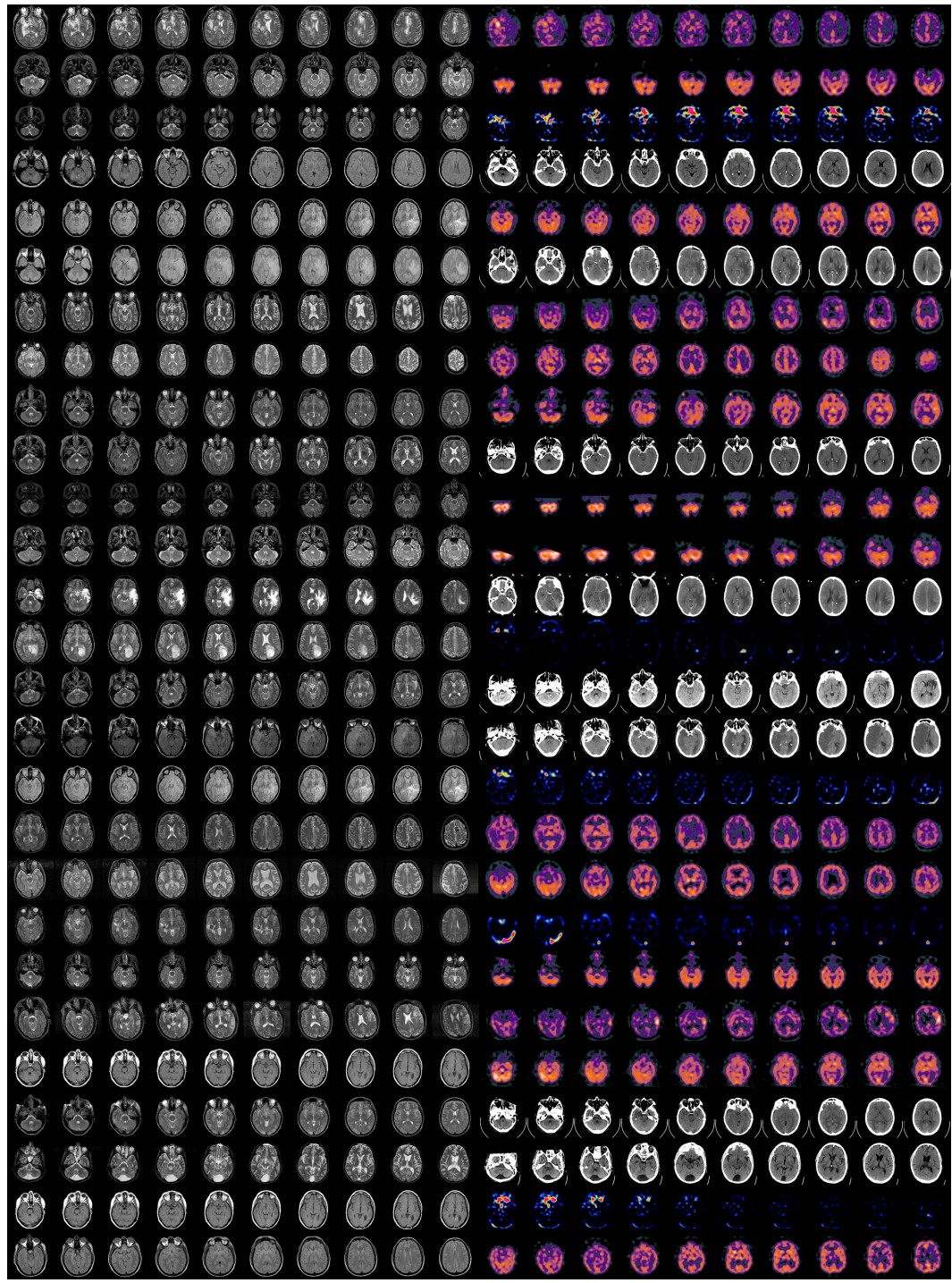

Figure 13: Dataset visualizations for *Medical Video Fusion* branch in VF-Bench. Columns 1–10 correspond to MRI video sequences, while columns 11–20 correspond to their respective CT, PET or SPECT video sequences.

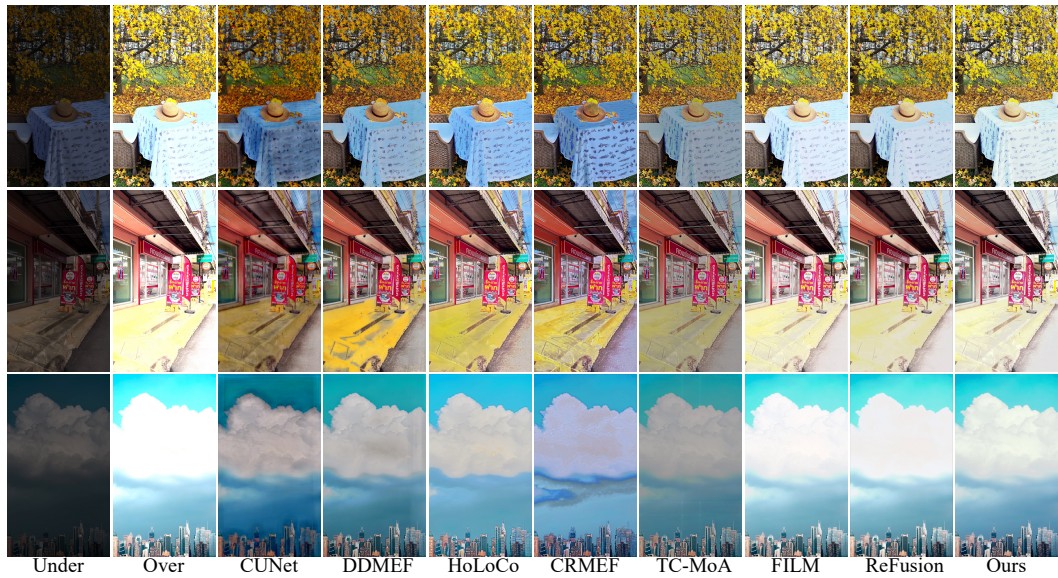

Under    Over    CUNet    DDMEF    HoLoCo    CRMEF    TC-MoA    FILM    ReFusion    Ours

Figure 14: Visualization comparison of the fusion results in the multi-exposure video fusion task.

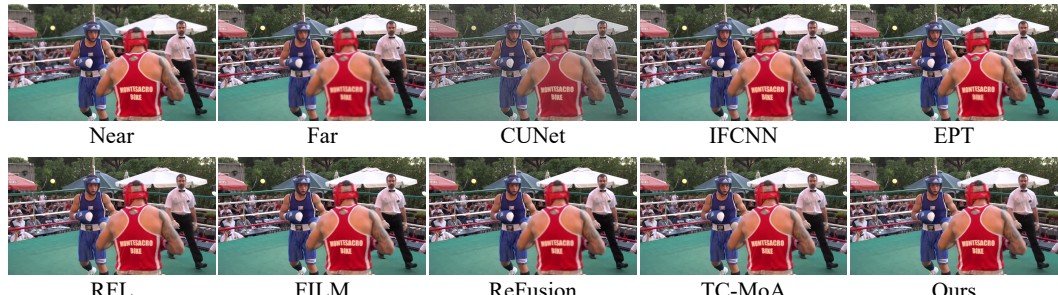

Figure 15: Visualization comparison of the fusion results in the multi-focus video fusion task.

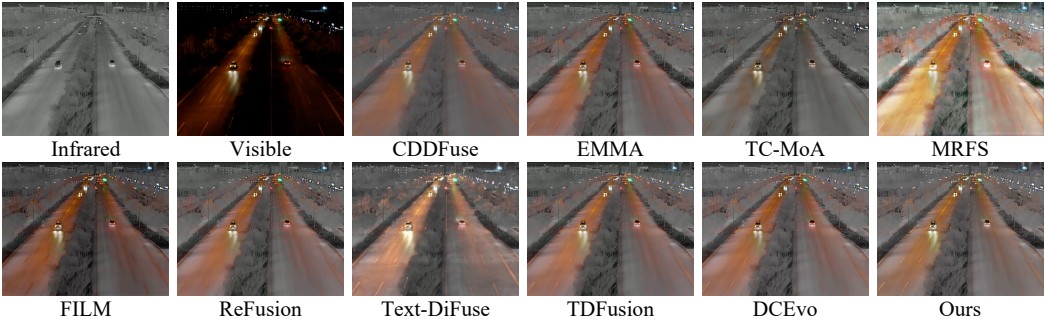

Figure 16: Visualization comparison of the fusion results in the infrared-visible video fusion task.

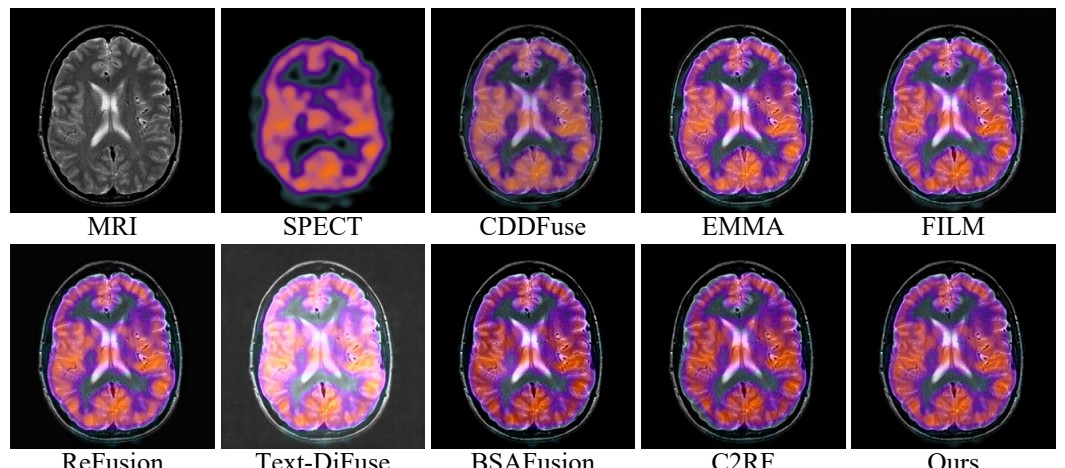

|       |       |         |      |      |
|-------|-------|---------|------|------|
| MRI   | SPECT | CDDFuse | EMMA | FILM |

| ReFusion | Text-DiFuse | BSAFusion | C2RF | Ours |
|----------|-------------|-----------|------|------|

Figure 17: Visualization comparison of the fusion results in the medical video fusion task.

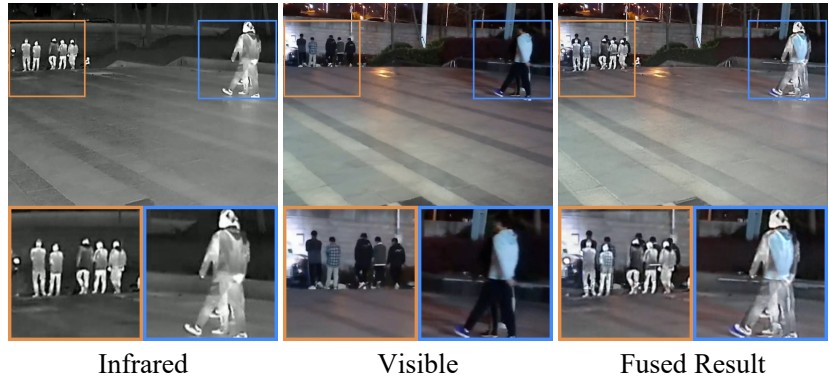

|          |         |              |
|----------|---------|--------------|
| Infrared | Visible | Fused Result |

Figure 18: Artifacts in fused video caused by misaligned input frames.

