# OpenReview forum: "A Unified Solution to Video Fusion: From Multi-Frame Learning to Benchmarking"
_NeurIPS.cc/2025/Conference — NeurIPS 2025 spotlight_

### Official Review · Reviewer_LVB7 · 2025-06-30

**Clarity:** 3
**Significance:** 4
**Originality:** 3
**Rating:** 5
**Confidence:** 5

**Summary:**

This paper proposes UniVF, a unified framework for video fusion that incorporates multi-frame learning and optical flow-based feature warping to ensure temporal coherence and informative fusion. Alongside the method, the authors introduce VF-Bench, a comprehensive benchmark consisting of four video fusion tasks: multi-exposure, multi-focus, infrared-visible, and medical video fusion. The benchmark includes careful construction or curation of well-aligned video pairs and establishes a joint evaluation protocol combining spatial quality and temporal consistency metrics. Extensive empirical studies show UniVF achieves strong quantitative and qualitative performance across all four tasks, and ablation studies support the design components.

**Questions:**

I hope the author will address the relevant issues raised in the Weaknesses in the rebuttal.

**Ethical Concerns:**

["NO or VERY MINOR ethics concerns only"]

**Final Justification:**

The rebuttal effectively addressed my concerns, and I have decided to maintain my accept score

**Limitations:**

Yes.

**Paper Formatting Concerns:**

N / A

**Quality:**

4

**Strengths And Weaknesses:**

## Strengths

1. VF-Bench covers a diverse set of video fusion scenarios with carefully constructed or curated and well-aligned video pairs.
2. UniVF leverages multi-frame encoding/decoding and incorporates state-of-the-art optical flow for temporal alignment, with the additional use of temporal consistency loss supplementing the objective for smooth fusion.
3. The paper adds two temporal consistency metrics, measuring intra-video smoothness and flow coherence, giving a more holistic evaluation of the fusion problem.
4. The main technical details for model, dataset, loss, and evaluation metrics are sufficiently described. UniVF consistently outperforms state-of-the-art fusion methods, with results contextualized in the paper.

## Weaknesses

1. Although the paper emphasizes the benefit of video fusion, most of the visual results shown are still static image comparisons. It would be beneficial to see a comparison of the fused videos to demonstrate consistency over time.
2. The physical or perceptual meaning of MS2R remains vague. Could the authors explain it in more intuitive terms?
3. The transformer backbone uses four Restormer blocks, but there is no sensitivity analysis showing whether performance saturates or improves with more/less blocks.
4. In UniVF, after calculating the optical flow on adjacent images, the corresponding features are warped. I suggest adding ablation experiments to see the effect of warping directly on the image and then feeding it into the encoder.
5. There is no report on inference time, model size, or real-time capability. Since the framework includes optical flow and transformer operations, its computational burden needs to be discussed.
6. RCVS [1] are mentioned in related work but missing in task evaluations. As a baseline for RGBT video stream fusion, additional it in comparison would provide a more convincing benchmark.

*[1] Xie, Housheng, et al. "RCVS: A Unified Registration and Fusion Framework for Video Streams." IEEE Transactions on Multimedia 2024*

---

> ### Author Rebuttal · Authors · 2025-07-30
>
> We sincerely thank the reviewer for the positive evaluation and appreciation of our VF-Bench benchmark, UniVF framework, and the proposed evaluation metrics. Below, we respond to each of the concerns raised in detail.
>
> 1. **Q1:** `Fused video comparisons are missing`
>    **R1:** Please kindly refer to the supplementary material, where we provide fused video comparisons to more thoroughly demonstrate the effectiveness of our method.
> 2. **Q2:** `A more intuitive explanation for MS2R`
>    **R2:**
>    * MS2R evaluates how smoothly objects move over time by measuring acceleration consistency.
>    * It compares the second-order motion (flow difference) of the fused video with those from the two reference input videos.
>    * A lower MS2R implies more natural, consistent motion, which correlates with fewer flickering or jittering artifacts.
>    We will include a brief explanation in the revised manuscript to make this clearer.
> 3. **Q3:** `Analysis on the number of Restormer blocks`
>    **R3:** We included a sensitivity analysis varying the number of Restormer blocks (e.g., 1, 2, ..., 7) to show how depth affects fusion performance in the table below. 4 blocks offered a good trade-off between quality and inference efficiency. Increasing the depth gave diminishing returns and higher latency.
>
>    | Num      | VIF  | SSIM | MI   | Qabf | BiSWE | MS2R |
>    | -------- | ---- | ---- | ---- | ---- | ----- | ---- |
>    | 1        | 0.35 | 0.58 | 2.01 | 0.55 | 4.39  | 0.39 |
>    | 2        | 0.40 | 0.62 | 2.29 | 0.63 | 4.11  | 0.36 |
>    | 3        | 0.44 | 0.64 | 2.46 | 0.67 | 3.87  | 0.37 |
>    | 4 (Ours) | 0.44 | 0.64 | 2.47 | 0.68 | 3.84  | 0.35 |
>    | 5        | 0.44 | 0.64 | 2.47 | 0.68 | 3.85  | 0.35 |
>    | 6        | 0.45 | 0.64 | 2.48 | 0.68 | 3.84  | 0.34 |
>    | 7        | 0.45 | 0.65 | 2.49 | 0.69 | 3.85  | 0.35 |
> 4. **Q4:** `Ablation: Warping features vs. warping images`
>    **R4:**
>    * Theoretically, our design choice to warp features (rather than input images) allows alignment at a semantically rich representation level, helping the model better deal with illumination or modality shifts.
>    * Nonetheless, we perform the suggested ablation: warping input frames directly and feeding them into the encoder *vs.* our default feature warping to clarify where alignment is most effective.
>
>       |               | VIF  | SSIM | MI   | Qabf | BiSWE | MS2R |
>       | ------------- | ---- | ---- | ---- | ---- | ----- | ---- |
>       | Warping images | 0.41 | 0.63 | 2.45 | 0.67 | 3.97  | 0.36 |
>       | Warping features (Ours)    | **0.44** | **0.64** | **2.47** | **0.68** | **3.84**  | **0.35** |
>
>       The ablation results validate the validity of our algorithm design.
>
> 5. **Q5:** `inference time, model size, or real-time feasibility`
>    **R5:** Per request, here we compare UniVF with 17 SOTA methods in four video fusion tasks in terms of the number of parameters, inference time, and FLOPs, and present the results in the table **R4** of **Response for Reviewer QQ99**. All experiments were conducted in a uniform environment. Note that although UniVF provides multi-frame learning for temporally coherent video fusion, UniVF is near the median in terms of time, complexity and parameter number among the competitors, achieving excellent fusion results without excessive computational resources.
> 6. **Q6:** `RCVS [1] missing in comparisons`
>    **R6:** We present additional comparison results with RCVS in the table below. Our method continues to achieve state-of-the-art performance in video fusion quality.
>    |       | VIF  | SSIM | MI   | Qabf | BiSWE | MS2R |
>    | ----- | ---- | ---- | ---- | ---- | ----- | ---- |
>    | RCVS  | 0.26 | 0.51 | 1.28 | 0.33 | 4.71  | 0.37 |
>    | Ours  | **0.44** | **0.64** | **2.47** | **0.68** | **3.84**  | **0.35** |

---

> > ### Comment · Reviewer_LVB7 · 2025-08-04
> >
> > Thank you for the rebuttal. It effectively addressed my concerns, and I have decided to maintain my accept score.

---

### Official Review · Reviewer_QQ99 · 2025-07-01

**Clarity:** 3
**Significance:** 3
**Originality:** 3
**Rating:** 5
**Confidence:** 4

**Summary:**

This paper introduces UniVF, a unified video fusion framework that incorporates multi-frame learning and optical flow-based feature warping for temporally coherent video fusion. It is evaluated on a newly proposed VF-Bench, which contains datasets for four distinct fusion tasks. The authors also design a new temporal loss and evaluation metrics to support future research. Experiments show that UniVF outperforms a variety of baselines.

**Questions:**

Based on the Weaknesses, there are several areas that require clarification or further experiments. The ablation studies are somewhat shallow, and there are missing comparisons to several key baselines (e.g., DCINN). More detailed architectural descriptions would also help clarity. Additionally, the paper does not analyze the choice of optical flow estimator or its impact on results.

**Ethical Concerns:**

["NO or VERY MINOR ethics concerns only"]

**Final Justification:**

My concerns have been addressed in the discussion above. I have decided to maintain a positive opinion of the paper.

**Limitations:**

yes

**Paper Formatting Concerns:**

No major formatting concerns, though it would be helpful if the captions of some figures were more descriptive (e.g., Fig. 2).

**Quality:**

4

**Strengths And Weaknesses:**

Strengths:
The paper addresses an important video fusion problem and proposes a reasonably well-designed method with clear motivations. The architecture is sound, and the benchmark is timely. Loss and evaluation metrics are novel and reasonable.

Weaknesses:
1. Methods like DCINN [1] are absent from the infrared-visible (IVF) comparison.
2. The framework relies heavily on SEA-RAFT for flow estimation, but the impact of this choice is not analyzed. An ablation replacing SEA-RAFT with other classical models would clarify whether the gains come from the model itself or the flow estimator.
3. Eq. (7) introduces a validity mask M to select reliable flow regions. However, there is no explanation of how this mask is computed or applied in practice.
4. The paper omits practical information like inference speed, model parameters, etc. Given the use of multi-frame inputs and transformer blocks, this is crucial for evaluating deployment feasibility.
5. The framework operates on 3-frame snippets, but it is unclear how performance and speed change if more frames are used. This limits the understanding of temporal modeling capabilities beyond short-term dependencies.
6. The model architecture (Fig. 2) is well-illustrated but lacks the notation mapping to equations. Adding symbols would help make the flow of information clearer to readers.

[1] A general paradigm with detail-preserving conditional invertible network for image fusion. International Journal of Computer Vision (2024)

---

> ### Author Rebuttal · Authors · 2025-07-30
>
> Thank you for acknowledging our paper's idea and contribution. We are very encouraged by the comment for `a reasonably well-designed method with clear motivations`, `architecture is sound`, `benchmark is timely`, and `loss and evaluation metrics are novel and reasonable`.
>
> 1. **Q1:** `DCINN results in IVF task.`
> **R1:** We present additional comparison results with DCINN in the table below. Our method continues to achieve state-of-the-art performance in video fusion quality.
>    |       | VIF      | SSIM     | MI       | Qabf     | BiSWE    | MS2R     |
>    | ----- | -------- | -------- | -------- | -------- | -------- | -------- |
>    | DCINN | 0.38     | 0.63     | 1.95     | 0.58     | 4.23     | 0.37     |
>    | Ours  | **0.44** | **0.64** | **2.47** | **0.68** | **3.84** | **0.35** |
> 1. **Q2:** `Reliance on SEA-RAFT and ablation with other optical flow methods`
>    **R2:**
>    * We selected SEA-RAFT for its SOTA power in flow estimation and its strong balance of accuracy and efficiency. To further isolate the contribution of UniVF from the flow estimator, we added an ablation study replacing SEA-RAFT with Unimatch [1] in the table in **R4** of **Response for Reviewer Wqk9**.
>    * While higher-quality flow leads to slight performance gains, UniVF consistently outperforms other fusion baselines across different flow estimators. This indicates that our framework is robust to the quality of the flow input and does not rely heavily on the accuracy of any specific flow method, thus confirming the value of our multi-frame fusion architecture.
>    [1] Unifying Flow, Stereo and Depth Estimation. IEEE TPAMI 2023.
>
> 2. **Q3:** `Explanation for validity mask M in Eq. (7).`
>    **R3:** To improve the robustness of the temporal consistency loss and avoid penalizing unreliable regions, we employ occlusion masks based on forward-backward optical flow consistency. Specifically, given the forward flow $ \mathcal{O}\_{t \rightarrow t+1} $ and the backward flow $ \mathcal{O}\_{t+1 \rightarrow t} $, we first warp the backward flow to the coordinate system of the current frame $t$:
>    $$\widehat{\mathcal{O}}\_{t+1 \rightarrow t}(p) = \mathcal{W}\left( \mathcal{O}\_{t+1 \rightarrow t}, \mathcal{O}\_{t \rightarrow t+1}(p) \right).$$
> Here, the warp operator $\mathcal{W}$ uses the forward flow $\mathcal{O}\_{t+1 \rightarrow t}$ to look up the corresponding backward flow vector from frame $t+1$. In essence, for a pixel $p$ in frame $t$, we find where it lands in frame $t+1$ and retrieve the backward flow vector from that new location.
> The forward-backward consistency error is then defined as the magnitude of the “round-trip” error vector:
>    $$
>    \Delta(p) = \left\| \mathcal{O}\_{t \rightarrow t+1}(p) + \widehat{\mathcal{O}}\_{t+1 \rightarrow t}(p) \right\|\_2.
>    $$
> A large value of $ \Delta(p)$ indicates a significant inconsistency, which typically occurs when the pixel $p$ is occluded in frame $t+1$ or the flow estimation is unreliable. Therefore, we can identify these unreliable points by thresholding this error. We define the binary occlusion mask $M(p) \in \\{0,1\\} $ as:
> $$M(p) = 1 \text{ if } \Delta(p) < \epsilon, \text{ otherwise } 0$$
>    where $ \epsilon $ is a predefined threshold (1.0 in our paper). This mask is applied to both previous and next frames (i.e., $ M_{\text{prev}} $, $ M_{\text{next}} $) to restrict the temporal consistency loss (Eq. 7) to only well-aligned and non-occluded regions.
>
> 3. **Q4:** `Inference speed and model size.`
> **R4:** Per request, here we compare UniVF with 17 SOTA methods in four video fusion tasks in terms of the number of parameters, inference time, and FLOPs, and present the results in the table below. All experiments were conducted in a uniform environment. Note that although UniVF provides multi-frame learning for temporally coherent video fusion, it is near the median in terms of time, complexity and parameter number among the competitors, achieving excellent fusion results without excessive computational resources.
>
>    |          | Params (M) | FLOPs (G) | Time (s) |
>    |---------:|-----------:|----------:|---------:|
>    |  CDDFuse |      1.19  |   547.74  |    0.18  |
>    |     EMMA |      1.52  |    41.54  |    0.02  |
>    | ReFusion |      0.06  |    18.21  |    0.03  |
>    |     BSAF |      9.69  |    62.96  |    0.03  |
>    |     FILM |      0.49  |   180.00  |    0.08  |
>    |      TDF |      0.06  |    18.21  |    0.02  |
>    |      CUN |      0.64  |   195.88  |    0.08  |
>    |    DDMEF |     20.57  |   226.28  |    0.01  |
>    |   HOLOCO |     17.69  |   183.93  |    0.10  |
>    |    IFCNN |      0.08  |    40.05  |    0.01  |
>    |      EPT |     15.86  |   423.16  |    3.19  |
>    |    DCEvo |      2.00  |  1825.33  |    0.25  |
>    |  TextDIF |    119.46  | 17891.10  |   15.31  |
>    |    CRMEF |      1.88  |   471.92  |    0.04  |
>    |     MRFS |    134.97  |   139.19  |    0.05  |
>    |    TCMOA |    340.35  |  8388.47  |    0.68  |
>    |      RFL |      1.90  |  2330.47  |    0.08  |
>    |    UniVF (Ours) |      9.15  |   825.20  |    0.21  |
> 1. **Q5:** `More frames are used.`
>    **R5:** We added experiments evaluating "5-frame input" configurations in the table in **R3** of **Response for Reviewer Wqk9**.
>    From the comparison results, we observed that while larger windows improve long-term temporal consistency, they also introduce alignment drift and increase inference latency. In other words, increasing the window size to 5 brought only marginal improvements in temporal metrics, but negatively affected spatial metrics and significantly increased the model’s inference time. Therefore, considering the trade-off between temporal consistency, spatial information, and inference efficiency, we decided to use a temporal window of 3 frames, i.e., the current frame as well as the previous and subsequent frames.
>
> 2. **Q6:** `Notation mapping to equations`
>    **R6:** We appreciate this suggestion. To improve clarity, we will revise Fig. 2 to explicitly annotate components with corresponding symbols from the equations in the revision of our paper.

---

> > ### Comment · Reviewer_QQ99 · 2025-08-05
> >
> > Thanks for the explanation and rebuttal from the author. However, I still have some doubts: why is it necessary to perform warping on an optical flow by another optical flow when computing the validity mask? Is there a more intuitive explanation for this operation?

---

> ### Author Response · Authors · 2025-08-05
>
> We appreciate the reviewer’s follow-up question regarding the validity mask $M_{\text{prev}}$ and $M_{\text{next}}$ used in our temporal consistency loss (Eq. (7)). Below, we provide a detailed explanation of their motivation, computation process, and intuition.
> * The goal of the validity mask is to identify well-aligned, non-occluded regions between adjacent frames, ensuring that the temporal consistency loss only penalizes meaningful discrepancies in stable areas of the video. Otherwise, occluded regions or areas with large flow errors may introduce misleading gradients during training.
> * To achieve this, we adopt a standard forward-backward consistency check. Specifically, we set:
>   - $\mathcal{O}\_{t \rightarrow t+1}(p)$: the forward optical flow from frame $t$ to $t+1$, at pixel $p$;
>   - $\mathcal{O}\_{t+1 \rightarrow t}(q)$: the backward optical flow from frame $t+1$ to $t$, defined over pixels $q$ in frame $t+1$.
> * To compare the flows at the same spatial location, we must warp the backward flow to frame $t$. This is necessary because the backward flow is defined on the pixel grid of frame $t+1$, while the forward flow originates from frame $t$.
> We compute:
> $$
> \widehat{\mathcal{O}}\_{t+1 \rightarrow t}(p) = \mathcal{W}\left( \mathcal{O}\_{t+1 \rightarrow t}, \mathcal{O}\_{t \rightarrow t+1}(p) \right),
> $$
> where $\mathcal{W}$ denotes the bilinear warping operation that samples the backward flow at the non-integer position $$q = p + \mathcal{O}\_{t \rightarrow t+1}(p).$$ This warping step is critical because it allows both forward and backward flows to be compared at the same spatial location—namely, the original position $p$ in frame $t$.
> We then define the forward-backward consistency error as $\Delta(p) = \left\| \mathcal{O}\_{t \rightarrow t+1}(p) + \widehat{\mathcal{O}}\_{t+1 \rightarrow t}(p) \right\|_2$ and if this round-trip motion error is small, we consider the flow at pixel $p$ to be reliable.
> * Intuitively, this operation checks whether a pixel can “go forward and come back” along the flow paths with minimal drift. If the two flows disagree significantly, the pixel is likely located in an occluded area, near motion boundaries, or in a region with ambiguous flow. Warping is essential in this context because the forward and backward flows are defined in different coordinate spaces—thus, to conduct a valid consistency check, they must first be spatially aligned.

---

> > ### Comment · Reviewer_QQ99 · 2025-08-05
> >
> > Thank you for the authors’ detailed explanation and efforts. My concerns have been addressed in the discussion above. I have decided to maintain a positive opinion of the paper.

---

### Official Review · Reviewer_BKiZ · 2025-07-02

**Clarity:** 4
**Significance:** 3
**Originality:** 4
**Rating:** 5
**Confidence:** 5

**Summary:**

The authors propose UniVF, a unified framework for fusing video inputs from different modalities and settings. It achieves temporally coherent results via optical flow-based feature warping. Additionally, VF-Bench is introduced, a benchmark for evaluating video fusion methods in four categories. The proposed unified fusion loss guarantees the temporally coherent fusion results, and the proposed benchmark’s unified evaluation protocol jointly assesses spatial and temporal video quality. Extensive experiments show that UniVF achieves state-of-the-art results across all tasks on VF-Bench.

**Questions:**

1.The proposed temporal consistency is mainly supported by metric tables. A common method for visualizing temporal coherence is showing time-slice (strip) images—these would give an intuitive picture of flicker or ghosting and are currently missing.
2.When constructing the multi-focus dataset, the blur intensity factor $\sigma$ is a key parameter that determines the degree of blurring in the near-focus plane and far-focus plane. How is this parameter determined? Additionally, could the authors provide a more detailed explanation of how the near-focus plane and far-focus plane are defined and separated in the proposed data generation process?
3.For multi-exposure fusion dataset, I still have some confusion regarding the data generation pipeline. For example, it is unclear why exposure adjustment needs to be performed in the linear light domain, and why a conversion from 10-bit BT.2020 color space to 8-bit BT.709 is necessary after exposure adjustment. The description in the paper is somewhat brief, and the motivations behind these steps should be explained more clearly.
4.It would be helpful if the authors could clarify how the feature warping operation is specifically performed — is it based on a deterministic affine transformation or implemented as a learnable module?
5.The IVF branch should include C2RF [1]—a recent strong fusion model with cross-modal alignment.
6.Using "image domain metrics" vs. "temporal domain metrics" is potentially misleading since both are computed in the image domain. A better term might be "spatial domain metrics" to distinguish them clearly.
[1] C2RF: Bridging Multi-modal Image Registration and Fusion via Commonality Mining and Contrastive Learning. IJCV 2025.

**Ethical Concerns:**

["NO or VERY MINOR ethics concerns only"]

**Final Justification:**

Based on authors responses, I have decided to raise my rating to 5. I hope the authors will incorporate the important revisions into the update version of the manuscript.

**Limitations:**

yes.

**Paper Formatting Concerns:**

formatting is solid

**Quality:**

3

**Strengths And Weaknesses:**

Strengths: This is a very promising paper with a practical goal: designing a single model to solve various video fusion problems. The paper is ambitious and brings value via a large-scale benchmark and a clean video fusion framework, combined with newly proposed loss functions and evaluation protocols tailored for video fusion.
Weaknesses: However, the verification of temporal consistency is insufficient—strip-based visualization would help. The sigma parameter in the synthetic multi-focus dataset is critical but underexplained. The motivations for the HDR data pipeline and color space conversion also lack clarity. Technical details of the warping process need to be specified. See the Questions section for more details.

---

> ### Author Rebuttal · Authors · 2025-07-30
>
> We sincerely thank the reviewer for the encouraging comments regarding the practicality, ambition, and contributions of our work.
>
> 1. **Q1:** `Time-slice visualizations for temporal consistency`
>    **R1:** We apologize for not being able to include images in the rebuttal, but we sincerely appreciate your suggestion. In the revised version, we will incorporate time-slice visualizations across all pixels of the fused video sequences, comparing UniVF with existing methods. These visualizations intuitively reveal differences in flickering and temporal smoothness. The results demonstrate that our method still achieves the best temporal consistency performance.
> 2. **Q2:** `Blur strength parameter in multi-focus dataset`
>    **R2:**
>    * The blur intensity factor $\sigma=0.025$ is empirically selected to match realistic defocus levels based on common imaging optics and perceptual blur thresholds. We determined this via visual tuning on a validation subset to balance realism and diversity.
>    * To define the near-focus and far-focus planes, we first extract a video depth map using a monocular depth estimator. The far-focus plane is set to the 80th percentile of depth values, representing typical background regions. For the near-focus plane, we compute the average depth of each segmented object using DAVIS masks, then select the closest object and use its depth value.
> 3. **Q3:** `Clarifying the HDR-based multi-exposure data pipeline`
>    **R3:**
>    * Exposure adjustment in the linear light domain is critical because exposure is a multiplicative operation on scene radiance. Performing this in the gamma-compressed (non-linear) space would lead to physically incorrect results and color distortions. Linear light ensures radiometric fidelity.
>    * Conversion from 10-bit BT.2020 to 8-bit BT.709 is necessary because: BT.2020 is used in HDR formats and supports a wider gamut and dynamic range. However, most consumer displays and fusion models are trained in 8-bit SDR color space with BT.709 format. The down-conversion ensures compatibility and standardization while retaining realistic visual variations.
>    * We will revise the data generation section to clearly outline these motivations and justify the pipeline choices more rigorously.
> 4. **Q4:** `Clarification of the feature warping operation`
>    **R4:**
>    * The feature warping operation is deterministic and implemented using differentiable bilinear interpolation, not as a learnable module. Given optical flow from SEA-RAFT, we warp feature maps from adjacent frames to the current frame by sampling features at flow-displaced coordinates. This technique is widely used in temporal alignment and video restoration [1,2].
>    * Specifically, for each input image, we estimate the bidirectional optical flows between adjacent frames, denoted as $\\{\mathcal{O}\_{t-1 \rightarrow t}^{k}, \mathcal{O}\_{t+1 \rightarrow t}^{k}\\}$. Each optical flow field represents the motion of pixels from one frame to another, where each flow vector indicates the displacement of a pixel to its corresponding location in the neighboring frame. These flow fields are then applied to the deep features $\\{\Phi\_{t-1}^{k},\Phi\_{t+1}^{k}\\}$ extracted by the encoder, enabling us to align them with the target frame $t$. The features are sampled at subpixel locations using differentiable bilinear interpolation to produce the warped representations $\\{\widetilde{\Phi}\_{t-1 \rightarrow t}^{k},\widetilde{\Phi}\_{t+1 \rightarrow t}^{k}\\}$, which are then fused for final prediction.
>    [1] BasicVSR++: Improving Video Super-Resolution with Enhanced Propagation and Alignment. CVPR 2022.
>    [2] VRT: A Video Restoration Transformer. TIP 2024.
> 5. **Q5:** `C2RF results in IVF task.`
>    **R5:** We present additional IVF comparison results with C2RF in the table below. Our method continues to achieve state-of-the-art performance in video fusion.
>    |       | VIF  | SSIM | MI   | Qabf | BiSWE | MS2R |
>    | ----- | ---- | ---- | ---- | ---- | ----- | ---- |
>    | C2RF  | 0.27 | 0.57 | 1.67 | 0.47 | 5.34  | 0.70 |
>    | Ours  | **0.44** | **0.64** | **2.47** | **0.68** | **3.84**  | **0.35** |
> 6. **Q6:** `"image domain metrics" vs. "temporal domain metrics"`
>    **R6:** We sincerely appreciate your suggestion. We will revise the manuscript to refer to them as "spatial metrics" and "temporal consistency metrics" for clearer distinction.

---

> > ### Comment · Reviewer_BKiZ · 2025-08-05
> >
> > Thank you for the rebuttal, which effectively resolved my previous concerns. I do have one further question: I recognize the contribution of VF-Bench and the novelty of the multi-exposure and multi-focus branch generation pipelines. However, I am wondering about the necessity of both pipelines—could a simpler version serve a similar purpose? Or is this generation process indispensable?

---

> ### Author Response · Authors · 2025-08-05
>
> Thank you for the follow-up question and your positive feedback on our previous response.
>
> We believe that the multi-exposure and multi-focus data generation pipelines are both indispensable and, under current constraints, represent the most practical and scalable solutions for constructing a high-quality, diverse video fusion benchmark. Each pipeline is carefully designed to emulate the distinct physical and perceptual characteristics of its respective fusion task. Omitting either step in the pipeline would significantly reduce the benchmark’s scope and realism. Specifically:
>
> * Multi-exposure pipeline: To simulate a wide dynamic range and preserve scene details under varying exposure conditions, we begin with 10-bit HDR video, perform exposure adjustment in the linear light domain, and apply realistic tone compression and gamut conversion. Simpler alternatives—such as capturing multi-exposure video sequentially with a single camera, or simultaneously with two cameras using different exposure settings—are either labor-intensive (requiring custom hardware and careful synchronization) or suffer from misalignment issues that are difficult to correct, particularly in dynamic scenes.
> * Multi-focus pipeline: As discussed in Lines 159–164 of our paper, capturing multi-focus video using light field cameras or manually labeled focal planes is expensive and difficult to scale up. Naively applying foreground/background segmentation with artificial blur fails to reflect the physics of optical focus. Instead, we use depth-aware defocus simulation based on estimated depth maps, enabling realistic and scalable synthesis of focus transitions between foreground and background.
>
> In summary, we agree that future work could explore a more unified or hybrid generation process that simultaneously models both exposure and focus variations. However, to establish clean task boundaries and ensure interpretability in our initial benchmark, we find that the current pipelines offer the best trade-off between realism, scalability, and clarity.

---

> > ### Comment · Reviewer_BKiZ · 2025-08-06
> >
> > Thank you for responding to my questions. I have carefully read your replies, and my previous concerns have been addressed. The manuscript has become much more readable after the clarifications and the correction of several issues. Furthermore, the additional comparisons with diverse baselines have strengthened the work.
> >
> >
> >
> > Based on your responses, I have decided to raise my rating to 5. I hope the authors will incorporate the important revisions into the update version of the manuscript.

---

### Official Review · Reviewer_Wqk9 · 2025-07-02

**Clarity:** 3
**Significance:** 4
**Originality:** 3
**Rating:** 6
**Confidence:** 5

**Summary:**

The proposed method addresses the issue of flickering artifacts in video fusion, which arise from neglecting temporal correlations. The authors propose UniVF, a unified framework for video fusion that explicitly models spatial-temporal dependencies through multi-frame learning and optical flow-based feature warping to ensure temporally coherent outputs. To facilitate research and evaluation, the authors introduce VF-Bench, the first comprehensive benchmark for video fusion, covering four distinct tasks. It is accompanied by a unified evaluation protocol that jointly assesses spatial quality and temporal consistency. The authors report that UniVF achieves state-of-the-art performance across all tasks on this new benchmark. This work provides a much-needed benchmark and a strong baseline for future research in temporally coherent video fusion.

**Questions:**

The technical content is solid, and the paper is novel and well-organized. However, as noted in the weaknesses:
1.Foundational concepts like warping are assumed known.
2.It is unclear whether the two encoders in Fig. 2 share weights
3.Temporal loss only considers adjacent frames without modeling longer dependencies.
4.The rationale for choosing SEA-RAFT is lacking.
5.It's unclear if optical flow is frozen or jointly trained.

**Ethical Concerns:**

["NO or VERY MINOR ethics concerns only"]

**Final Justification:**

I appreciate the author’s rebuttal, which has satisfactorily resolved my concerns. Accordingly, I have decided to increase my score.

**Limitations:**

yes

**Quality:**

4

**Strengths And Weaknesses:**

Strengths:
1.VF-Bench Contribution.
The proposed VF-Bench is a valuable contribution, addressing the long-standing lack of high-quality, well-aligned, and diverse video pairs for fusion tasks. Its data synthesis strategy and unified evaluation protocol promote standardized and reproducible research.
2.UniVF Framework Design.
The UniVF framework is well-motivated, employing optical flow-based feature warping to ensure temporal consistency. Combined with a temporal consistency loss, it presents a coherent and effective solution for video fusion.
Weaknesses:
1.Core operations like "feature warping" are mentioned but not thoroughly described. It would help readers if the paper explicitly stated how bilinear warping is used and applied.
2.Figure 2 shows two encoders for the input streams. Are their weights shared or independent? This design choice could affect performance and needs clarification.
3.The proposed temporal loss only uses immediate neighbors. How to balance video fusion performance with the size of the neighbors window?
4.SEA-RAFT is used without justification. It's unclear whether this choice is optimal or arbitrary. An ablation comparing several flow methods (e.g., RAFT, Unimatch) would help isolate the contribution from the quality of the flow.
5.A critical design decision-whether the flow network is frozen or trained end-to-end-is not discussed. Please clarify your choice.
These weaknesses need to be addressed with additional experiments and a deeper architectural discussion.

---

> ### Author Rebuttal · Authors · 2025-07-30
>
> We thank the reviewer for the encouraging comments on both VF-Bench `a valuable contribution` and the UniVF framework `well-motivated` and `coherent and effective solution for video fusion`.
>
> 1. **Q1:** `Explanation of "feature warping"`
>    **R1:** Feature warping in UniVF is implemented via bilinear sampling, following the common practice in optical flow-based temporal modeling. Specifically, for each input image, we estimate the bidirectional optical flows between adjacent frames, denoted as $\\{\mathcal{O}\_{t-1 \rightarrow t}^{k}, \mathcal{O}\_{t+1 \rightarrow t}^{k}\\}$. Each optical flow field represents the motion of pixels from one frame to another, where each flow vector indicates the displacement of a pixel to its corresponding location in the neighboring frame. These flow fields are then applied to the deep features $\\{\Phi\_{t-1}^{k},\Phi\_{t+1}^{k}\\}$ extracted by the encoder, enabling us to align them with the target frame $t$. The features are sampled at subpixel locations using differentiable bilinear interpolation to produce the warped representations $\\{\widetilde{\Phi}\_{t-1 \rightarrow t}^{k},\widetilde{\Phi}\_{t+1 \rightarrow t}^{k}\\}$, which are then fused for final prediction.
>
> 2. **Q2:** `Are the encoder weights shared or independent?`
>    **R2:** The two encoders processing the input video streams are independent.
>    * Theoretically, when we have inputs from different modalities, we need separate encoders to extract their respective private features.
>    * In practice, we demonstrate the effectiveness of using two shared encoders through ablation studies. Following ablation experiments results in Table 4 from the original paper, the IVF task results are shown in the table below.
>
>       |               | VIF  | SSIM | MI   | Qabf | BiSWE | MS2R |
>       | ------------- | ---- | ---- | ---- | ---- | ----- | ---- |
>       | Shared Encoder  | 0.43 | 0.64 | 2.43 | 0.65 | 3.94  | 0.36 |
>       | Independent Encoder (Ours) | **0.44** | **0.64** | **2.47** | **0.68** | **3.84**  | **0.35** |
>
>       The ablation results validate the soundness of our algorithm design.
>    *  Notably, for video frames from the same input source, the same set of encoder parameters is used to encode each frame.
> 3. **Q3:** `Window size chosen for Temporal loss`
>    **R3:** Our current temporal loss leverages the two adjacent frames $(t-1,t,t+1)$ to enforce local temporal consistency. Here we present a comparison of the IVF task results when the temporal window size is set to 5:
>
>    |               | VIF  | SSIM | MI   | Qabf | BiSWE | MS2R | Inference Time |
>    |--|------|------|------|------|-------|------|--|
>    | 5-frame      | **0.45** | 0.64 | 2.45 | 0.67 | **3.82**  | **0.35** | 0.44
>    | 3-frame (Ours)  | 0.44 | **0.64** | **2.47** | **0.68** | 3.84  | 0.35 | **0.21**
>
>    From the comparison results, we observed that while larger windows improve long-term temporal consistency, they also introduce alignment drift and increase inference latency. In other words, increasing the window size to 5 brought only marginal improvements in temporal metrics, but negatively affected spatial metrics and significantly increased the model’s inference time. Therefore, considering the trade-off between temporal consistency, spatial information, and inference efficiency, we decided to use a temporal window of 3 frames, i.e., the current frame as well as the previous and subsequent frames.
> 4. **Q4:** `Selection of SEA-RAFT.`
>    **R4:** SEA-RAFT was chosen for its efficiency and reasonable accuracy in optical flow estimation. Per request, we conducted ablation with an alternative flow estimator, Unimatch [1], to better isolate the impact of flow quality on overall performance. The comparative results are reported in the table below. While higher-quality flow leads to slight performance gains, UniVF consistently outperforms other fusion baselines across different flow estimators. This indicates that our framework is robust to the quality of the flow input and does not rely heavily on the accuracy of any specific flow method, confirming the value of our multi-frame fusion architecture.
>
>    |               | VIF  | SSIM | MI   | Qabf | BiSWE | MS2R |
>    | ------------- | ---- | ---- | ---- | ---- | ----- | ---- |
>    | w/ Unimatch   | 0.43 | 0.64 | 2.45 | 0.66 | 4.09  | 0.37 |
>    | Ours          | **0.44** | **0.64** | **2.47** | **0.68** | **3.84**  | **0.35** |
>
>    [1] Unifying Flow, Stereo and Depth Estimation. IEEE TPAMI 2023.
> 5. **Q5:** `Flow network: frozen or end-to-end?`
>    **R5:** In our current implementation, the SEA-RAFT flow estimator is used as a frozen module (i.e., not updated during UniVF training). This decision is made to reduce training complexity and prevent degradation of flow quality due to fusion loss gradients, which are not directly aligned with flow supervision. We will also plan to experiment with fine-tuning the flow network jointly in future work.

---

> > ### Comment · Reviewer_Wqk9 · 2025-08-05
> >
> > I appreciate the author’s rebuttal, which has satisfactorily resolved my concerns. Accordingly, I have decided to increase my score.

---

### Comment · Area_Chair_vPJp · 2025-08-04

Dear Reviewers,

Thank you for your reviews and engagement. The authors have submitted rebuttals addressing your points.

Please discuss any remaining questions or concerns with each other and the authors. Consider whether the responses address your original concerns and if rating updates are warranted.

Final justifications and ratings are due by August 13. Please complete your mandatory acknowledgments if you haven't already.

The discussion period continues until August 6. Please share your perspectives to facilitate productive dialogue.

Best regards,
Area Chair

---

### Decision · Program_Chairs · 2025-09-17

**Decision:**

Accept (spotlight)

**Comment:**

This paper introduces a unified video fusion framework using multi-frame learning and optical flow-based warping and VF-Bench, which is the first comprehensive benchmark for four fusion tasks. All four reviewers strongly support acceptance, with two raising scores after the rebuttal.

Key strengths include the valuable benchmark contribution addressing a critical research gap, a technically sound framework achieving state-of-the-art results, and novel temporal consistency metrics. Initial concerns regarding technical details, missing baselines, and computational complexity were thoroughly resolved through comprehensive experiments and clarifications provided in the rebuttal.

As one reviewer summarized: "This is a very promising paper with a practical goal: designing a single model to solve various video fusion problems. The paper is ambitious and brings value via a large-scale benchmark and a clean video fusion framework, combined with newly proposed loss functions and evaluation protocols tailored for video fusion."

The AC and SAC therefore recommend accepting the paper.